# POODLE: Improving Few-shot Learning via Penalizing Out-of-Distribution Samples

**Duong H. Le** [*]
VinAI Research
v.duonglh5@vinai.io

**Khoi D. Nguyen** [*]
VinAI Research
khoinguyenucd@gmail.com

**Khoi Nguyen**
VinAI Research
ducminhkhoi@gmail.com

**Quoc-Huy Tran**
Retrocausal, Inc.
huy@retrocausal.ai

**Rang Nguyen**
VinAI Research
rangnhm@gmail.com

**Binh-Son Hua**
VinAI Research &
VinUniversity

## Abstract

In this work, we propose to leverage out-of-distribution samples, *i.e.,* unlabeled samples coming from outside target classes, for improving few-shot learning. Specifically, we exploit the easily available out-of-distribution samples (*e.g.,* from base classes) to drive the classifier to avoid irrelevant features by maximizing the distance from prototypes to out-of-distribution samples while minimizing that to in-distribution samples (*i.e.,* support, query data). Our approach is simple to implement, agnostic to feature extractors, lightweight without any additional cost for pre-training, and applicable to both inductive and transductive settings. Extensive experiments on various standard benchmarks demonstrate that the proposed method consistently improves the performance of pretrained networks with different architectures. Our code is available at https://github.com/VinAIResearch/poodle.

## 1 Introduction

Learning with limited supervision is a key challenge to translate the research efforts of deep neural networks to real-world applications where large-scale annotated datasets are prohibitively costly to acquire. This issue has motivated the recent topic of few-shot learning (FSL), which aims to build a system that can quickly learn new tasks from a small number of labeled data.

A popular group of methods in FSL focus on strengthening the backbone network by various techniques, from increasing model capacity [6, 11], self-supervised learning (SSL) [18, 51, 65], to knowledge distillation (KD) [53]. With these techniques, few-shot methods are expected to learn better representations that are more robust and generalized. However, even if the network can discover visual features and semantic cues, few-shot learners have to deal with a key challenge - the *ambiguity*: as we have only a small amount of support evidence, there are multiple plausible hypotheses at the inference stage. Existing works, therefore, rely on the developed *inductive bias* of the network (during the pretraining stage), such as shape bias [47, 14], to reduce the hypothesis space.

In this work, we view the classification problem as conditional reasoning, *i.e., "if X has P then X is Q"*. Human beings are good at learning such inferences, thus quickly grasping new concepts with minimal supervision. More importantly, humans learn new concepts in context - where we have already had prior knowledge about other entities. According to *mental models* in cognitive science, when assessing the validity of an inference, one would retrieve counter-examples, *i.e.,* which do not

---

[*]First two authors contribute equally.

35th Conference on Neural Information Processing Systems (NeurIPS 2021).

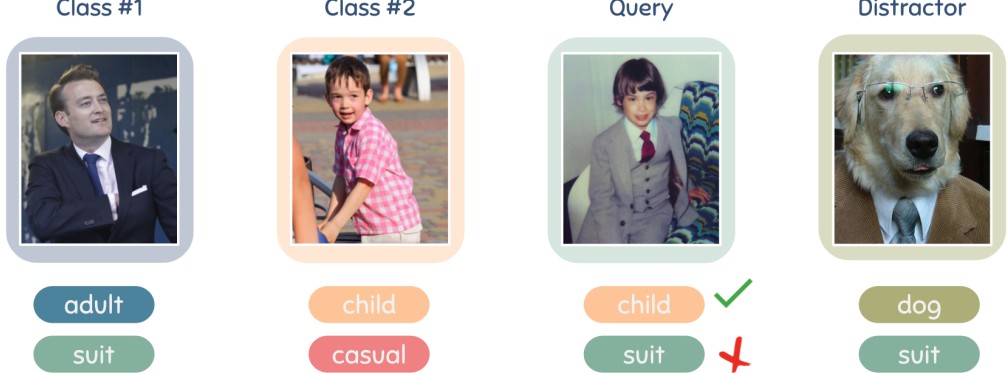

Figure 1: Illustration of advantages of counter-example data. The query image has features from both support classes (*i.e.,* a child and a suit), which makes classification ambiguous. The prediction result would depend on the inductive bias or prior knowledge of the network. By using an out-of-distribution sample, it becomes clear that the query result should favor Class 2.

lead to the conclusion despite satisfying the premise [9, 13, 29, 30, 55]. Thus, if there exists at least one of such counter-examples, the inference is known to be erroneous.

Hence, we attempt to equip few-shot learning with the above ability so that it can eliminate incorrect hypotheses when learning novel tasks in a data-driven manner. Specifically, we leverage out-of-distribution data, *i.e.,* samples belonging to classes separated from novel tasks [2], as counter-examples for preventing the learned prototypes from overfitting to their noisy features. To that end, when learning novel tasks, we adopt the *large margin principle* in metric learning [59] to encourage the learned prototypes to be close to support data while being distant from out-of-distribution samples.

Our approach is complementary to existing works in FSL and could be combined to advance the state of the art. Moreover, our method is agnostic to the backbone network; thus, it does not have the need of a training phase to adopt as in SSL and KD, while incurring just a little overhead at inference (for fine-tuning prototypes with approximately 200 gradient updating steps).

In summary, our contributions are as follows:

- We propose a novel yet simple approach to learn the inductive bias of deep neural networks for FSL by leveraging out-of-distribution data. We empirically show that out-of-distribution data only require weak labels (*i.e.,* in the form of whether a sample is in- or out-of-distribution) even in challenging problems such as cross-domain FSL.

- We introduce a new loss function to implement the above idea, which is applicable for both inductive and transductive inference. Our extensive experiments on different standard benchmarks show that the proposed approach consistently improves the performance of various network architectures.

- We validate the effectiveness of our method in various FSL settings, including cross-domain FSL.

## 2   Related work

Over the past few years, considerable amount of research efforts [15, 16, 28, 34, 44, 56, 50, 64] have been invested in FSL. They can roughly be classified into two main categories: *optimization-based* and *metric-based* approaches. Optimization-based methods [15, 16, 28, 34, 44] seek a meta-learner that can quickly adjust the parameters of another learner to a new task, given only a few support images. In particular, [35, 15] propose learning to initialize a classifier whose parameters can be obtained with a small number of gradient updates on the novel classes. Metric-based approaches [56, 50, 64] learn a task-agnostic embedding space for measuring the similarity between images. For example, Matching Networks [56] utilizes a weighted nearest neighbor classifier, while Prototypical Networks [50] uses the mean features of support images as the prototype for each class. Recently, DeepEMD [64] adopts the Earth Mover's distance to compute the distance between image patches as their distance.

---

[2]We use OOD samples and distractor samples interchangeably

To further boost the performance, recent works have incorporated additional techniques such as *self-supervised learning* and *knowledge distillation* during training, and *transductive inference* during testing, which we will review in the following:

**Self-supervised learning.** The goal of self-supervised learning is to learn representations from unlabeled data. For FSL, this could be achieved by combining the supervised main task with a self-supervised pretext task which includes predicting image rotations [19], predicting relative patch locations [12], or solving jigsaw puzzles [40]. A few works incorporating self-supervised learning for FSL have been developed recently, e.g., Gidaris et al. [18] suggest pre-training the embedding network by using a combination of supervised and self-supervised loss, *i.e.,* predicting image rotations.

**Knowledge distillation.** Knowledge distillation is a machine learning technique that seeks to compress the knowledge contained in a larger model (teacher) into a smaller one (student). It was introduced by Bucilua et al. [4] and later adopted to deep learning by Hinton et al. [23]. Recently, Tian et al. [53] show that incorporating self-distillation into FSL can boost the performance by $1 - 2\%$.

**Transductive inference.** Transductive inference aims to leverage the query set (which can be seen as unlabeled data during inference) in addition to the labeled support set. Transductive approaches [39, 24, 43, 66, 2, 10] significantly outperform their inductive counterparts, which do not exploit the query set. For example, TPN [39] constructs a graph whose nodes are support and query images and propagates labels from the support to query images, while CAN [24] utilizes confidently classified query images as part of the support set. Recently, TIM [2] assumes a uniform distribution of novel classes, which is often the case for the current FSL benchmarks, and exploits that assumption for boosting the performance. Furthermore, [15, 52] can be considered as transductive methods since the information from query data is used for batch normalization.

In this paper, we improve the performance of the classifier on novel classes by introducing a novel loss function, which penalizes out-of-distribution samples. Our method is complementary to the above approaches and can be combined to establish a new state-of-the-art for FSL. As the time of camera-ready, we are aware of a concurrent work [8] that also leverages the distractor samples to refine the classifier in FSL. Contrast to that work, our loss function is inherently applicable to both inductive and transductive inference. Furthermore, we also discover the effectiveness of uniform random features, which obviates the need for accessible OOD samples for in-domain FSL.

## 3  Preliminary

We first define some notations used in the following sections. Let $(\mathbf{x}, y)$ consist of a sample image $\mathbf{x}$ and its corresponding class label $y$. Let $D_b = \{(\mathbf{x}_i, y_i)\}_{i=1}^{N_b}$ denote the labeled base samples used for feature pre-training. Next, we denote $D_s = \{(\mathbf{x}_i, y_i)\}_{i=1}^{N_s}$ and $D_q = \{(\mathbf{x}_i, y_i)\}_{i=1}^{N_q}$ as the labeled support samples and query samples respectively. Note that the labels for the query samples are only used for evaluation purposes. The support samples and query samples belong to the novel classes $C_n$, which are separated from the base classes $C_b$, *i.e.,* $C_b \cap C_n = \varnothing$. In few-shot learning, we aim to learn a classifier that exploits support data to predict labels for query samples. We use a pre-trained feature extractor, usually kept fixed, to produce input to the classifier.

In this work, we consider fine-tuning the classifier on the support set only (*inductive learning*) and on both the support and query set (*transductive learning*). We also consider both *in-domain* and *cross-domain* FSL. In the former, the novel classes $C_n$ and the base classes $C_b$ are from the same domain (*e.g.,* images from Image-Net), while for the latter, $C_n$ and $C_b$ are from different domains and the domains of $C_b$ and $C_n$ are referred to as the source and target domains, respectively.

**Pretrained feature extractor.** Let $f_\theta$ be the feature extractor trained on the base data with the standard cross-entropy loss, which we also refer as the **"simple baseline"**. We also seek to strengthen the baseline with orthogonal techniques such as self-supervised learning (SSL) [18, 51] and knowledge distillation (KD) [53]. With SSL, we employ the **"rot baseline"**, which is trained with the standard cross-entropy loss and an auxiliary loss to predict the rotation angles of the perturbed images. We further apply the born-again strategy [17] for the *rot baseline* in two generations to construct the **"rot + KD baseline"**. More details of our baselines can be found in Section A.

**Novel tasks inference.** In the few-shot scenario, we freeze the feature extractor $f_\theta$ and train a classifier for each task. Let $\mathbf{W} = [\mathbf{w}_0, \cdots, \mathbf{w}_k]^\top \in \mathbb{R}^{K \times D}$ be the weight matrix of the classifier,

where $D$ is the dimension of the encoded vector (output by the feature extractor) and $K$ is the number of classes (*i.e.,* number of ways) for each task. The predictive distribution over classes $p(k|\mathbf{x}_i, \mathbf{W})$ is given by:

$$p(k|\mathbf{x}_i, \mathbf{W}) = \frac{\exp(-\gamma \cdot d(\mathbf{z}_i, \mathbf{w}_k))}{\sum_j \exp(-\gamma \cdot d(\mathbf{z}_i, \mathbf{w}_j))}, \quad \text{where} \quad \mathbf{z}_i = \frac{f_\theta(\mathbf{x}_i)}{\|f_\theta(\mathbf{x}_i)\|_2} \tag{1}$$

where $\gamma$ is the learnable scaling factor [42] and $d(\cdot, \cdot)$ denotes the distance function. We use the squared Euclidean distance in our experiments unless otherwise mentioned. In our implementation, we initialize each weight vector $\mathbf{w}_k$ to the mean of the sample features in the support set $\mathbf{w}_k = \frac{1}{|\mathcal{S}_k|} \sum_{\mathbf{x} \in \mathcal{S}_k} f_\theta(\mathbf{x})$ with $\mathcal{S}_k$ being the support set of the $k^{\text{th}}$ class similar to Prototypical Network [38].

**Inductive bias.** Inductive learning is a process of learning a general principle by observing specific examples. Given limited support data, it is possible to have multiple explanations on the query data, with each corresponding to a different prediction. Inductive bias allows the learner to systematically favor one explanation over another, rather than favoring a model that overfits to the limited support data. Figure 1 shows an ambiguous classification example that can be explained by multiple hypotheses. Two decision rules possibly learnt from the support data are: 1) People in suit belong to class 1; and 2) Boys belong to class 2.

Both rules are simple (satisfying Occam's razor), and can be used to classify the query sample. However, it is unclear which rule the learner would favor; it is only possible to know after training. Hence, solely learning with support data can be obscure. One way to narrow down the hypothesis space is using counter-examples to assess the inductive validity [29, 9]. In Figure 1, the out-of-distribution (OOD) sample hints that the suit should be considered irrelevant, and hence the rule 1 should be rejected.

## 4 Proposed approach

We introduce our novel technique for few-shot learning namely **P**enalizing **O**ut-**O**f-**D**istribution samp**LE**s (POODLE). Specifically, we attempt to regularize few-shot learning and improve the generalization of the learned prototypes by leveraging prior knowledge of in- and out-of-distribution samples. Our definition is as follows. *Positive samples* are in-distribution samples provided in the context of the current task that includes both support and query samples. *Negative samples*, in contrast, do not belong to the context of the current task, and hence are out-of-distribution. Negative samples can either provide additional cues that reduce ambiguity, or act as distractors to prevent the learner from overfitting. Note that negative samples should have the same domain as positive samples so that their cues are insightful to the learner, but positive and negative samples are not required to have the same domain as the base data.

To effectively use positive and negative samples in few-shot learning, the following conditions must be met: 1) The regularization guided by out-of-distribution samples can be combined with traditional loss functions for classification; 2) The regularization should be applicable for both inductive and transductive inference; and 3) The requirement on negative data should be minimal *i.e.,* does not need any sort of labels except for aforementioned conditions.

### 4.1 Refining prototype with distractor samples

We formulate the regularization as a new objective function for training. To capitalize negative samples to reduce the ambiguity, we propose leveraging the large-margin principle as in *Large Margin Nearest Neighbors* (LMNN) [59, 59]. In these works, Weinberger *et al.* propose a loss function with two competing terms to learn a distance metric for nearest neighbor classification: a *"pull"* term to penalize the large distance between the embeddings of two nearby neighbors, which likely belong to the same class, and a *"push"* term to penalize the small distance between the embeddings of samples of difference classes.

In this work, we seek to learn prototypes for all categories instead of a distance metric. We do not have labels (*i.e.,* categories) of all samples (*e.g.,* transductive inference) but only the prior knowledge about whether a sample is *in-* or *out-*of-distribution. Thus, we adapt the above objective of margin maximization for these two groups, namely in- and out-of-distribution, with the distance function being the *sum* of distances from a sample to *all* prototypes. The goal is minimizing distances from positive samples to prototypes, while maximizing distances from negative samples to prototypes.

In our objective function, we keep the original "pull" term while introducing our new "push" term: we do not explicitly enforce large distances between positive samples and negative samples, but only attempt to maximize distances between prototypes and negative samples:

$$\mathcal{L}_{naive} = \sum_{i=1}^{N_{pos}} \sum_{k=1}^{K} \gamma \cdot d(\mathbf{w}_k, \mathbf{x}_i) - \sum_{j=1}^{N_{neg}} \sum_{k=1}^{K} \gamma \cdot d(\mathbf{w}_k, \mathbf{x}_j) \tag{2}$$

Note that $\gamma$ is scaling factor of distance-based classifier as in Equation 1. However, this objective does not take into account class assignments for positive samples. To tackle this problem, we use weighted distances between prototypes and samples to simultaneously optimize both objectives:

$$\mathcal{L}_{margin} = \underbrace{\sum_{i=1}^{N_{pos}} \sum_{k=1}^{K} \gamma \cdot d(\mathbf{w}_k, \mathbf{x}_i) \mathcal{S}_G[\, p(k|\mathbf{x}_i, \mathbf{W})]}_{\mathcal{L}_{pull}} - \underbrace{\sum_{j=1}^{N_{neg}} \sum_{k=1}^{K} \gamma \cdot d(\mathbf{w}_k, \mathbf{x}_j) \mathcal{S}_G[\, p(k|\mathbf{x}_j, \mathbf{W})]}_{\mathcal{L}_{push}} \tag{3}$$

where $\mathcal{S}_G[\cdot]$ is the stop-gradient operator. Intuitively, the positive sample will *"pull"* the prototypes to its location proportional to its distance to the prototypes. Subsequently, the prototype of each class will move closer to the positive samples of that class. At the same time, the prototype is enforced to move away from the negative samples, thus discarding features that might lead to high similarity to out-of-distribution data.

We note that removing $\mathcal{S}_G[\cdot]$ in Equation 3 would result in a different underlying objective, which empirically leads to a decrease in performance. Particularly, the objective without stop-gradient will also be compounded of an auxiliary term for entropy maximization (see Section B). Thus, devoid of meticulous regularization would deteriorate the performance. The above observation is in line with Boudiaf *et al.* [2]. In summary, POODLE optimizes the classifier on novel tasks with the following objectives:

$$\boxed{\mathcal{L}_{\text{POODLE}} = \mathcal{L}_{ce} + \alpha \cdot \mathcal{L}_{pull} - \beta \cdot \mathcal{L}_{push}} \tag{4}$$

where $\alpha$ and $\beta$ control the "push" and "pull" coefficients respectively and $\mathcal{L}_{ce}$ denotes the standard cross-entropy loss. To the best of our knowledge, the proposed objective is the first loss function (in test phase) that can be effective for both inductive and transductive inference. Please see the supplemental document for the pseudo-code (Section C).

## 4.2 On negative samples

**Choice of negative samples.** For *in-domain* FSL, we simply leverage the base data as negative samples. For *cross-domain* FSL, one approach would be using a set of samples drawn from classes that are disjoint to novel classes $C_n$ as negative examples. However, in some cases such negative examples might not be available. Thus, we consider another approach where we have a set of unlabeled data of a set of classes $C_u$ from the target domain ($C_u$ might overlap with $C_n$) as negative examples, similar to [41]. We name the above two approaches as *disjoint* and *noisy* negative sampling, respectively in the context of cross-domain FSL. Intuitively, when the number of categories of unlabeled data (*i.e.,* $|C_u|$) grow larger, the noise of mixing positive and negative samples in *noisy* negative samples pool will be reduced.

**The burden of additional data.** It is worth pointing out that POODLE does not break the setup of FSL as it does not require any additional data of novel classes, thus, still has a few samples from novel tasks. As POODLE obligates additional data in the form of OOD samples, one might ask how practical the algorithm is. For in-domain FSL, the negative samples can be easily obtained by adopting the training data as we already know $C_n \cap C_b = \varnothing$. Even in extreme case where we only have access to the pretrained models, we empirically find that ($\ell_2$-normalized) uniformly random noise can work surprisingly well (Section 4.3). For cross-domain FSL, even though POODLE requires additional OOD data, we find that these samples can be: 1) noisy with both positive and negative samples; and 2) fairly efficient - in our experiments, we can "reuse" 400 OOD samples for all tasks, which is very efficient compared to other methods that might use up to 20% *unlabeled* data of training set [41].

## 4.3 Intriguing effectiveness of uniform random features

One caveat of POODLE is the obligation of accessing to OOD samples. Fortunately, we find that POODLE can use the uniform random features on the hypersphere as negative samples **for in-domain**

**FSL**. Particularly, we can uniformly sample latent vectors from $\mathbb{S}^{D-1}$ as an alternative to the distractor samples from base training data, thus, eliminating the need for accessing to training data.

Our use of uniform distribution as negative examples is inspired by that feature uniformity is a desirable property for contrastive loss [57, 5], and so a good representation prefers such uniformity. The problem of uniformly distributing points on the unit hypersphere is related to minimizing pairwise loss [31, 1], which links well to the theory of supervised classification with softmax and cross-entropy (equivalent to minimizing pairwise loss or maximizing mutual information) [3, 45].

Therefore, using uniform distribution as negative samples works for in-domain FSL because they will approximate the features of samples from base training data. For cross-domain FSL, this approach will fail because random features (which are similar to samples from the training domain) have a large discrepancy with the target domain, thus, not inducing meaningful cues. Intuitively, the more similar between domains of OOD and test samples, the higher performance gain POODLE can achieve. Our experiments in Section 5.4 empirically prove aforementioned postulation.

## 5 Experiments

In this section, we conduct extensive experiments to demonstrate the performance gain of our method on standard inductive, transductive, cross-domain FSL. To demonstrate the robustness of our method across datasets/network architectures, **we keep the hyperparameters fixed for all experiments.**

### 5.1 Experimental setup

**Datasets.** We evaluate our approach on three common FSL datasets. The *mini*-Imagenet dataset [56] consists of 100 classes chosen from the ImageNet dataset [48] including 64 training, 16 validation, and 20 test classes with 60,000 images of size $84 \times 84$. The *tiered*-Imagenet [46] is another FSL dataset which is also derived from the ImageNet dataset with 351 base, 97 validation, and 160 test classes with 779,165 images of size $84 \times 84$. Caltech-UCSD Birds (CUB) has 200 classes split into 100, 50, 50 classes for train, validation and test following [6]. Furthermore, we also carry out experiments on iNaturalist 2017 (iNat) [54], EuroSAT [22], and ISIC-2018 (ISIC) [7] for the (extreme) cross-domain FSL. The description for these datasets can be found in the supplemental document (Section D.1).

**Implementation details.** We use ResNet12 as our feature extractor. It is a residual network [61] with 12 layers split into 4 residual blocks. For pre-training on the base classes, we train our backbones with the standard cross-entropy loss for 100 epochs. The optimizer has a weight decay of $5e^{-4}$, and the initial learning rate of 0.05 is decreased by a factor of 10 after 60, 80 epochs in mini-ImageNet and 60, 80, 90 epochs in tiered-ImageNet. We use the batch size of 64 for all the networks. For fine-tuning on the novel classes, we utilize Adam optimizer [32] with fixed learning rate of 0.001, $\beta_1 = 0.9$, $\beta_2 = 0.999$, and do not use weight decay. The classifier is trained with 250 iterations. The coefficients of "push/pull" loss are $\alpha = 1$ and $\beta = 0.5$ respectively. For negative samples, we randomly select $K = 400$ samples from the out-of-distribution samples pool for each novel task. We evaluate the performance of POODLE in 5-way-1-shot and 5-way-5-shot settings on 2000 random tasks with 15 queries each.

### 5.2 Standard FSL

We first experiment with the standard FSL setup (also known as in-domain FSL) in which the number of query samples is uniformly distributed among classes.

**Evaluation with various baselines.** Table 1 shows the results of our approach with various baselines, which described in Section 3, with the inductive inference (positive samples are support images). As can be seen, our approach consistently boosts the performance of all baselines by a large margin (1-3%). Interestingly, the POODLE-R variant outperforms the CE loss variant significantly and performs comparatively with the POODLE-B variant. We hypothesize that the normalized representations of the samples extracted from the base classes are uniformly distributed.

Table 2 demonstrates the efficacy of each loss term of POODLE in transductive inference (positive samples are support and query images). We can see that using the "pull" loss with query samples

Table 1: Comparison to different baselines for the standard FSL (a.k.a in-domain FSL) on *mini*-ImageNet, *tiered*-Imagenet and CUB in the **inductive** setting with *Resnet-12* as backbone. Here, POODLE-B and POODLE-R indicate that the negative samples are sampled from the base classes and the random uniform distribution respectively.

| Baseline | Variant | *mini*-ImageNet | | *tiered*-ImageNet | | CUB | |
|---|---|---|---|---|---|---|---|
| | | 1-shot | 5-shot | 1-shot | 5-shot | 1-shot | 5-shot |
| Simple | CE Loss | 61.33 | 80.76 | 67.42 | 84.44 | 75.60 | 89.88 |
| | POODLE-B | 63.79 | **81.41** | 69.26 | 84.97 | 75.96 | **90.06** |
| | POODLE-R | **64.38** | 81.35 | **69.34** | **85.03** | **76.22** | 89.99 |
| Rot | CE Loss | 64.57 | 82.89 | 68.26 | 85.09 | 76.39 | 91.10 |
| | POODLE-B | 66.78 | **83.65** | 69.74 | 85.45 | 77.26 | 91.30 |
| | POODLE-R | **67.20** | 83.54 | **69.86** | **85.56** | **77.35** | **91.43** |
| Rot + KD | CE Loss | 65.91 | 82.95 | 69.43 | 84.93 | 79.70 | 92.28 |
| | POODLE-B | 67.50 | **83.71** | 70.42 | **85.26** | **80.23** | 92.36 |
| | POODLE-R | **67.80** | 83.50 | **70.47** | 85.24 | 80.05 | **92.37** |

Table 2: Comparison to different baselines on *mini*-ImageNet, *tiered*-Imagenet and CUB with and without the proposed loss in the **transductive** settings. Here, $\mathcal{L}_{pull}^{b}$ and $\mathcal{L}_{pull}^{u}$ indicate that the negative samples are sampled from the base classes and the random uniform distribution respectively.

| Baseline | Loss | *mini*-ImageNet | | *tiered*-ImageNet | | CUB | |
|---|---|---|---|---|---|---|---|
| | | 1-shot | 5-shot | 1-shot | 5-shot | 1-shot | 5-shot |
| Simple | CE | 61.33 | 80.76 | 67.42 | 84.44 | 75.60 | 89.88 |
| | $CE + \mathcal{L}_{pull}$ | 70.91 | 83.04 | 75.58 | 86.10 | 84.91 | 91.60 |
| | $CE + \mathcal{L}_{pull} - \mathcal{L}_{push}^{b}$ | 74.21 | **83.71** | 78.72 | 86.57 | 87.04 | **91.84** |
| | $CE + \mathcal{L}_{pull} - \mathcal{L}_{push}^{u}$ | **74.82** | 83.67 | **79.24** | **86.64** | **87.13** | 91.60 |
| Rot | CE | 64.57 | 82.89 | 68.26 | 85.09 | 76.39 | 91.10 |
| | $CE + \mathcal{L}_{pull}$ | 74.42 | 85.20 | 76.59 | 86.75 | 85.93 | 93.02 |
| | $CE + \mathcal{L}_{pull} - \mathcal{L}_{push}^{b}$ | 77.30 | **85.91** | 79.74 | 87.25 | 88.68 | 93.23 |
| | $CE + \mathcal{L}_{pull} - \mathcal{L}_{push}^{u}$ | **77.69** | 85.86 | **80.31** | **87.33** | **89.07** | **93.35** |
| Rot + KD | CE | 65.91 | 82.95 | 69.43 | 84.93 | 79.70 | 92.28 |
| | $CE + \mathcal{L}_{pull}$ | 74.90 | 85.09 | 77.07 | 86.50 | 88.22 | 93.68 |
| | $CE + \mathcal{L}_{pull} - \mathcal{L}_{push}^{b}$ | **77.56** | **85.81** | 79.67 | **86.96** | 89.88 | **93.80** |
| | $CE + \mathcal{L}_{pull} - \mathcal{L}_{push}^{u}$ | 77.24 | 85.53 | **79.97** | 86.91 | **90.02** | 93.68 |

improve the inductive baseline significantly, being as effective as other transductive algorithms. Combining with "push" term, the classifier is further enhanced.

We also conduct experiments with additional loss functions including self-supervised loss (SSL) and knowledge distillation (KD) in Figure 2a. Our approach consistently improves the generalization of all baselines without additional computation cost (in the training phase) as justified in Figure 2b. The improvement of POODLE on other backbones and datasets is reported in the supplemental document (Section D.2).

**Comparison to the state-of-the-art approaches.** We report the performance of our network in comparison with state-of-the-art methods in both transductive and inductive settings (with and without information from the query images) in Table 3. We can see that our approach remarkably improves the performance of the baseline and achieves a comparable performance with the state-of-the-art approaches in the tiered-ImageNet. In mini-ImageNet and CUB we significantly outperform the prior work in both inductive and transductive settings. Experimental results on other backbones are reported in the supplementary document (Section D.2).

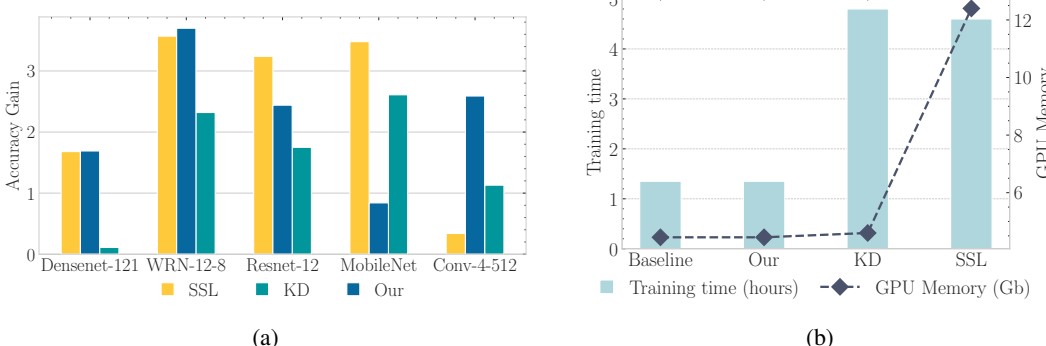

(a)                                                    (b)

Figure 2: (a) Effectiveness of our approach applied to standard FSL. (a) Our approach consistently yields accuracy gains on different backbones, in comparison with those obtained by self-supervised loss (SSL) and knowledge distillation (KD) in 5-way-1-shot protocol on *mini*-Imagenet with **inductive** setting. (b) Total time and memory cost (in training stage) when adopting SSL, KD, and our method. Note that we apply KD for $T = 2$ generation as in [53]. No large overhead is incurred in our method.

Table 3: Comparison to the state-of-the-art methods on *mini*-ImageNet, *tiered*-Imagenet and CUB using **inductive** and **transductive** settings. The results obtained by our models (blue pearl-shaded) are averaged over 2,000 episodes.

| Method | Transd. | Backbone | *mini*-ImageNet | | *tiered*-ImageNet | | CUB | |
|---|---|---|---|---|---|---|---|---|
| | | | 1-shot | 5-shot | 1-shot | 5-shot | 1-shot | 5-shot |
| MAML [15] | | ResNet-18 | 49.6 | 65.7 | - | - | 68.4 | 83.5 |
| RelatNet [24] | | ResNet-18 | 52.5 | 69.8 | - | - | 68.6 | 84.0 |
| MatchNet [56] | | ResNet-18 | 52.9 | 68.9 | - | - | 73.5 | 84.5 |
| ProtoNet [50] | | ResNet-18 | 54.2 | 73.4 | - | - | 73.0 | 86.6 |
| Neg-cosine [37] | ✗ | ResNet-18 | 62.3 | 80.9 | - | - | 72.7 | 89.4 |
| MetaOpt [33] | | ResNet-12 | 62.6 | 78.6 | 66.0 | 81.6 | - | - |
| SimpleShot [58] | | ResNet-18 | 62.9 | 80.0 | 68.9 | 84.6 | 68.9 | 84.0 |
| Distill [53] | | ResNet-12 | 64.8 | 82.1 | **71.5** | **86.0** | - | - |
| Rot + KD + POODLE | | ResNet-12 | **67.80** | **83.72** | 70.42 | 85.26 | **80.23** | **92.36** |
| RelatNet + T [24] | | ResNet-12 | 52.4 | 65.4 | - | - | - | - |
| TPN [39] | | ResNet-12 | 59.5 | 75.7 | - | - | - | - |
| TEAM [43] | | ResNet-18 | 60.1 | 75.9 | - | - | - | - |
| Ent-min [10] | | ResNet-12 | 62.4 | 74.5 | 68.4 | 83.4 | - | - |
| CAN+T [24] | ✓ | ResNet-12 | 67.2 | 80.6 | 73.2 | 84.9 | - | - |
| LaplacianShot [66] | | ResNet-18 | 72.1 | 82.3 | 79.0 | 86.4 | 81.0 | 88.7 |
| TIM-GD [2] | | ResNet-18 | 73.9 | 85.0 | **79.9** | **88.5** | 82.2 | 90.8 |
| Simple + POODLE | | ResNet-12 | 74.21 | 83.71 | 78.72 | 86.57 | 87.04 | 91.84 |
| Rot + KD + POODLE | | ResNet-12 | **77.56** | **85.81** | 79.67 | 86.96 | **89.93** | **93.78** |

## 5.3   Cross-domain FSL

In this section, we conduct experiments to demonstrate the efficacy of our approach even with a challenging task: **extreme** cross-domain FSL as first introduced in [21, 41]. Many FSL algorithms are known to fail in such a challenging scenario [6].

Recall that we have two setups for cross-domain FSL (Section 4.2), in which cases, the negative samples are drawn from *disjoint* and *noisy* negative samples pool. Here, we provide the detailed setting of each scheme when transferring a network trained on *mini*-Imagenet to target domains. It is worth mentioning that we only consider *inductive* inference for comparison with prior work.

**Disjoint negative samples.** As mentioned before, we evaluate the results of cross-domain FSL on test split of these datasets while employing the train split to draw negative samples. Since the number of categories in EuroSAT and ISIC is relatively small (10 and 7 respectively) compare to the number

Table 4: The results of the domain-shift setting from *mini*-Imagenet to CUB, iNat, ISIC, and EuroSAT with *Rot + KD baseline*. The results obtained by our models (blue pearl-shaded) are averaged over 2,000 episodes. The baselines with ⋆ notation show the results using the setup of **disjoint** negative samples, and the baselines without the ⋆ show the results using the setup of **noisy** negative samples.

| | CUB | | iNat | | ISIC | | EuroSAT | |
| Baseline | 1-shot | 5-shot | 1-shot | 5-shot | 1-shot | 5-shot | 1-shot | 5-shot |
|---|---|---|---|---|---|---|---|---|
| Transfer [41] | - | - | - | - | 30.71 | 43.08 | 60.73 | 80.30 |
| SimCLR [41] | - | - | - | - | 26.25 | 36.09 | 43.52 | 59.05 |
| Transfer + SimCLR [41] | - | - | - | - | 32.63 | 45.96 | 57.18 | 77.61 |
| STARTUP (no SS) [41] | - | - | - | - | 32.24 | 46.48 | 62.90 | 81.81 |
| STARTUP [41] | - | - | - | - | 32.66 | **47.22** | 63.88 | **82.29** |
| Rot + KD + CE | 49.98 | 69.44 | 48.90 | 66.58 | 32.35 | 43.81 | 65.00 | 80.01 |
| Rot + KD + POODLE-R | 50.04 | 69.78 | 48.94 | 66.87 | 32.21 | 43.87 | 63.97 | 79.85 |
| Rot + KD + POODLE-B | **52.61** | **70.78** | **50.62** | **67.31** | **33.56** | 44.17 | **66.21** | 80.51 |
| Rot + KD + CE ⋆ | 50.28 | 69.64 | 48.80 | 66.72 | - | - | - | - |
| Rot + KD + POODLE-R ⋆ | 50.25 | 69.97 | 48.84 | 67.03 | - | - | - | - |
| Rot + KD + POODLE-B ⋆ | **53.11** | **70.96** | **50.46** | **67.45** | - | - | - | - |

of "way" in novel task (5), we only concern with iNat and CUB. Precisely, we follow [6] and [58] to split CUB and iNat respectively.

**Noisy negative samples.** We assume that we have $20\%$ unlabeled data from the target domain and the rest $80\%$ of data is used for testing. Although a large number of classes are beneficial to POODLE as discussed above, we also carry out experiments with ISIC and EuroSAT to evaluate the performance of **extreme** cross-domain FSL with our approach.

From the Table 4 we can see that the improvement of POODLE with the **disjoint** negative samples (the bottom rows) and the **noisy** negative samples (the middle rows) when transferring knowledge from mini-ImageNet to iNat, CUB, ISIC, and EuroSAT. We can observe that despite the extreme gap between target/source domains and a very noisy mix of in- and out-of-distribution samples in negative samples pool (in case of EuroSAT and ISIC), POODLE can successfully boost the performance of baselines in all experiments. We also report the performance of other approaches, which have same setup as **noisy** negative samples, in Table 4.

## 5.4 Ablation study

In this section, we present results of some presentative ablation study to get more insight of POODLE behaviors. For more in-depth experiments, we refer reader to supplementary document.

### 5.4.1 The effect of domain discrepancy between positive-negative samples

Intuitively, the more similar between domains of OOD and test samples, the higher performance gain POODLE can achieve. To understand how our method works when OOD data is from various domains, we train our classifier and compare the performance when using random uniform distribution and other datasets as negative examples. The result of this experiment is reported in Table 5. We can observer that the more similar of OOD domain to test domain, the higher the performance gain we can achieve. Thus, we should always aim to leverage the OOD samples of test domain.

**On the performance of random features:** For in-domain FSL *i.e.,* test domains are mini-Imagenet and tiered-Imagenet, using uniform examples has the best and second-best accuracy when tested on mini-ImageNet and tiered-ImageNet, respectively. As aforesaid, the uniform random features will reflect the distribution of mini-Imagenet samples (which the network is pre-trained on). Thus, random features are effective for in-domain FSL.

For cross-domain FSL (*i.e.,* test domains are CUB and EuroSAT), using random uniform features - which is approximately equal to using samples from mini-Imagenetet - does not work because the discrepancy between the source and target domain is extremely large, *e.g.,* animal (mini-Imagenet) vs satellite images (EuroSAT).

Table 5: Evaluating (simple) Resnet-12 trained on mini-Imagenet on different test/OOD domains in the 1-shot inductive protocol (10,000 episodes). *random, mini, tiered, CUB, EuroSAT* denote *mini*-Imagenet, *tiered*-Imagenet, CUB, EuroSAT dataset respectively. The OOD/test samples are drawn from the standard train/test split of each dataset, respectively. The 95% confidence interval is roughly 0.20 for all results.

| | | **OOD domain** | | | | | |
| | | *w/o OOD* | *random* | *mini* | *tiered* | *CUB* | *EuroSAT* |
|---|---|---|---|---|---|---|---|
| | *mini* | 61.63 | **64.30** | 64.08 | 63.72 | 62.71 | 61.76 |
| **Test domain** | *tiered* | 63.04 | 64.28 | 64.01 | **64.50** | 63.85 | 62.78 |
| | *CUB* | 48.55 | 48.88 | 49.01 | 49.10 | **51.40** | 49.18 |
| | *EuroSAT* | 65.18 | 63.85 | 64.58 | 64.25 | 64.70 | **66.04** |

Table 6: The results of the various approaches to learn a better classifier with pretrained *Rot + KD baseline* on *mini*-Imagenet. The results are obtained by averaging over 10,000 episodes.

| Method | 1-shot | 5-shot |
|---|---|---|
| baseline | $66.32 \pm 0.20$ | $82.99 \pm 0.13$ |
| 1) w/ learning cosine classifier | $66.66 \pm 0.20$ | $82.84 \pm 0.13$ |
| 2) w/ naive OOD | $66.36 \pm 0.20$ | $83.08 \pm 0.13$ |
| 3) w/ large-margin w/o negative samples | $66.32 \pm 0.20$ | $83.01 \pm 0.13$ |
| 4) w/ label smoothing | $66.13 \pm 0.20$ | $80.81 \pm 0.13$ |
| POODLE-B | $\mathbf{67.84 \pm 0.20}$ | $\mathbf{83.72 \pm 0.13}$ |
| POODLE-R | $\mathbf{68.20 \pm 0.20}$ | $\mathbf{83.60 \pm 0.13}$ |

### 5.4.2 Comparison against simple approaches for enhance pretrained models

To better understand the effectiveness of our method, we also compare to four methods that learn better classifiers on novel classes with pre-trained features. The results are reported in Table 6.

1. *Learning cosine classifier*: The classifier in inference phase is fine-tuned using CE loss with the support samples of each task. In our experiments, this approach does not bring any meaningful improvement similar to [58].

2. *Naive OOD*: we fine-tune a $(k + 1)$-way classifier with 1 additional class for OOD samples using CE loss. Naive OOD performs worse because the OOD samples are drawn from several classes of the training set, which are well-clustered and separated, we cannot find a "prototype" with a linear classifier to match all of them.

3. *Large-margin w/o negative samples*: only use push term in POODLE. This approach is not better than cosine-distance because the initialized prototype (mean of all samples from specific class) is already well-clustered and optimal *i.e.,* close to the ground-truth class prototype and far away from others for the observed samples.

4. *Label smoothing:* we use label smoothing for CE loss (ground-truth class has the probability of 0.9 and uniformly distribute 0.1 to the rest. This method does not work well because it only makes learned prototypes not be far away from samples from other classes.

## 6 Conclusions and future work

In this work, we have proposed the concept of leveraging out-of-distribution samples set to improve the generalization of few-shot learners and realize it by a simple yet effective objective function. Our approach consistently boosts the performance of FSL across different backbone networks, inference types (inductive/transductive), and the challenging cross-domain FSL.

Future work might seek to exploit different sampling strategies (*i.e.,* how to select negative samples) to further boost the performance and reduce time/memory complexity; another interesting direction is enhancing the robustness of the classifier when we have both positive and negative samples in the same sampling pool; leveraging domain adaptation to reduce the need of in-domain negative samples is also a promising research direction.

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
