# A  Baseline Implementation

**Simple Baseline.**   As stated above, the *simple baseline* is constructed by training the feature extractor on base class with standard cross-entropy loss, which yield the objective function:

$$\mathcal{L}^{simple} = \mathcal{L}_{clas}(\theta, \psi; D_b) = \frac{1}{N_b} \sum_{i=0}^{N_b} \sum_{j=0}^{|C_b|} y_{ij} \log p_\psi(j|f_\theta(\mathbf{x}_i)) \tag{5}$$

where $\theta$ and $\psi$ represent parameters of feature extractor and the linear classification, respectively.

**Rotation Baseline.**   For SSL, we implement the *"rotation baseline"* (*Rot baseline*). In detail, during global classification training, we rotate each image with **all** predefined angles: $\mathbf{x}_i^r = T(\mathbf{x}_i, r) \forall r \in \mathcal{R}$ with $\mathcal{R} = \{0°, 90°, 180°, 270°\}$ and enforce the encoder to recognize the correct translation. Thus, the combined objective function of this baseline can be defined as:

$$\mathcal{L}^{rot} = \mathcal{L}^{simple} + \lambda_{ssl}\mathcal{L}_{ssl}(\theta, \phi; D_b, \mathcal{R}) \tag{6}$$

Precisely, we utilize a seperate classifier $g_\phi$ on top of feature extractor $f_\theta$ to predict the pertubation of images via cross-entropy loss with four classes corresponding to four possible translations similar to Gidaris *et al.* [18]. Accordingly, the SSL loss function $\mathcal{L}_{ssl}$ is given by:

$$\mathcal{L}_{ssl}(\theta, \phi; D_b, \mathcal{R}) = \frac{1}{N_b|\mathcal{R}|} \sum_{i=0}^{N_b} \sum_{j \in \mathcal{R}} \sum_{k=0}^{|\mathcal{R}|} \mathbb{I}(k = j) \log p_\phi(k|f_\theta(\mathbf{x}_i^j)) \tag{7}$$

Here, $\mathbb{I}(\cdot)$ is the indicator function and $p_\phi(\cdot|f_\theta)$ denotes the (predicted) probability of rotation angle.

**Rotation + KD Baseline.**   The *"rotatiton + kd baseline"* (*Rot + KD baseline*) is constructed by combining knowledge distillation [23] with rotation classification. Explicitly, we first train the feature extractor following the configuration of *rotation baseline*, and then exert the born-again strategy [17] to perform knowledge distillation for $T$ generations. The loss function of this baseline is given by:

$$\mathcal{L}^{rot+kd} = \mathcal{L}^{rot} + \lambda_{kd\_clas}\mathcal{L}_{kd\_clas}(\theta_t, \psi_t; D_b, \theta_{t-1}, \psi_{t-1}, \tau) +$$
$$\lambda_{kd\_ssl}\mathcal{L}_{kd\_ssl}(\theta_t, \phi_t; D_b, \mathcal{R}, \theta_{t-1}, \phi_{t-1}, \tau) \tag{8}$$

The above objectives optimize $\mathcal{L}_{clas}$ and $\mathcal{L}_{ssl}$ with both groundtruth labels along with the prediction from teacher models *i.e.,* the model (of the same architecture) trained from the previous generation:

$$\mathcal{L}_{kd\_clas}(\theta_t, \psi_t; D_b, \theta_{t-1}, \psi_{t-1}, \tau) = \frac{1}{N_b} \sum_{i=0}^{N_b} \sum_{j=0}^{|C_b|} p_{\psi_{t-1}}(j|f_{\theta_{t-1}}(\mathbf{x}_i), \tau) \log p_{\psi_t}(j|f_{\theta_t}(\mathbf{x}_i), \tau) \tag{9}$$

$$\mathcal{L}_{kd\_ssl}(\theta_t, \phi_t; D_b, \mathcal{R}, \theta_{t-1}, \phi_{t-1}, \tau) = \frac{1}{N_b|\mathcal{R}|} \sum_{i=0}^{N_b} \sum_{j \in \mathcal{R}} \sum_{k=0}^{|\mathcal{R}|} p_{\phi_{t-1}}(k|f_{\theta_{t-1}}(\mathbf{x}_i^j), \tau) \log p_{\phi_t}(k|f_{\theta_t}(\mathbf{x}_i^j), \tau)$$
$$\tag{10}$$

Following Tian *et al.* [53], we distill the network for $T = 2$ generations and use the **last** checkpoint of previous generation as the teacher for next generation. For simplicity, we set the hyperparameters $\lambda_{ssl} = \lambda_{kd\_ssl} = \lambda_{kd\_clas} = 1$ and $\tau = 4$ for *all* experiments.

# B  On the importance of stop-gradient operator

In this section, we elaborate the bleak outcome when removing the stop-gradient operator of loss function in Equation 3. As discussed before, the soft-weight in *"pull/push"* loss terms should be used to guide the degree of force. Directly optimize these soft weights would lead to different underlying objective that **jointly maximizing conditional entropy** as we show below.

First, we define the objective function (for each positive/negative term) in Equation 3 as *weighted distance* loss:

$$\mathcal{L}_{wd} = \sum_{k=1}^{K} \gamma \cdot d(\mathbf{w}_k, \mathbf{x}_i) \mathcal{S}_G[\, p(k|\mathbf{x}_i, \mathbf{W})] \tag{11}$$

Furthermore we define the *log-sum-exp* loss function as:

$$\mathcal{L}_{lse} = -\log \sum_{k=1}^{K} \exp(-\gamma \cdot d(\mathbf{w}_k, \mathbf{x}_i)) \tag{12}$$

**Lemma B.1.** *The two loss function $\mathcal{L}_{wd}$ and $\mathcal{L}_{lse}$ have same set of solutions.*

*Proof.* This can be proved by taking the gradient of two terms *w.r.t.* $j^{\text{th}}$ prototype. Particularly, the gradient of $\mathcal{L}_{lse}$ *w.r.t.* $\mathbf{w}_j$ is given by:

$$\frac{\partial \mathcal{L}_{lse}}{\partial \mathbf{w}_j} = -\frac{\partial \log \sum_{k=1}^{K} \exp(-\gamma \cdot d(\mathbf{w}_k, \mathbf{x}_i))}{\partial \mathbf{w}_j} \tag{13}$$

$$= -\frac{1}{\sum_{k=1}^{K} \exp(-\gamma \cdot d(\mathbf{w}_k, \mathbf{x}_i))} \times \frac{\partial \sum_{k=1}^{K} \exp(-\gamma \cdot d(\mathbf{w}_k, \mathbf{x}_i))}{\partial \mathbf{w}_j} \tag{14}$$

$$= -\frac{\exp(-\gamma \cdot d(\mathbf{w}_j, \mathbf{x}_i))}{\sum_{k=1}^{K} \exp(-\gamma \cdot d(\mathbf{w}_k, \mathbf{x}_i))} \times \frac{\partial(-\gamma \cdot d(\mathbf{w}_j, \mathbf{x}_i))}{\partial \mathbf{w}_j} \tag{15}$$

$$= p(j|\mathbf{x}_i, \mathbf{W}) \cdot \frac{\partial(\gamma \cdot d(\mathbf{w}_j, \mathbf{x}_i))}{\partial \mathbf{w}_j} \tag{16}$$

On the other hand, the gradient of $\mathcal{L}_{wd}$ term *w.r.t.* to $\mathbf{w}_j$ is given by:

$$\frac{\partial \mathcal{L}_{wd}}{\partial \mathbf{w}_j} = \frac{\partial \sum_{k=1}^{K} \gamma \cdot d(\mathbf{w}_k, \mathbf{x}_i) \mathcal{S}_G[\, p(k|\mathbf{x}_i, \mathbf{W})]}{\partial \mathbf{w}_j} \tag{17}$$

$$= p(j|\mathbf{x}_i, \mathbf{W}) \frac{\partial(\gamma \cdot d(\mathbf{w}_j, \mathbf{x}_i))}{\partial \mathbf{w}_j} \tag{18}$$

Thus, both loss function should optimize the same underlying objective.

**Corollary B.1.1.** *Taking out the* stop-gradient *operator in "weighted distance" loss results in a objective that jointly minimizing conditional entropy of label given raw features and weighted distance (i.e., the old loss function).*

*Proof:* We refer to this objective function as *"Expected Distance"* loss. We can prove above corollary by rewriting the objective function as below:

$$\mathcal{L}_{ed} = \sum_{k=1}^{K} \gamma \cdot d(\mathbf{w}_k, \mathbf{x}_i)\, p(k|\mathbf{x}_i, \mathbf{W}) = -\sum_{k=1}^{K} p(k|\mathbf{x}_i, \mathbf{W}) \log\left[\exp\left(-\gamma \cdot d(\mathbf{w}_k, \mathbf{x}_i)\right)\right] \tag{19}$$

$$= -\sum_{k=1}^{K} \left[p(k|\mathbf{x}_i, \mathbf{W}) \log \frac{\exp(-\gamma \cdot d(\mathbf{w}_k, \mathbf{x}_i))}{\sum_{k'} \exp(-\gamma \cdot d(\mathbf{w}_{k'}, \mathbf{x}_i))} + p(k|\mathbf{x}_i, \mathbf{W}) \log \sum_{k'} \exp(-\gamma \cdot d(\mathbf{w}_{k'}, \mathbf{x}_i))\right] \tag{20}$$

$$= \underbrace{\left(-\sum_{k=1}^{K} p(k|\mathbf{x}_i, \mathbf{W}) \log p(k|\mathbf{x}_i, \mathbf{W})\right)}_{\text{conditional entropy loss}} - \underbrace{\log \sum_{k'} \exp\left(-\gamma \cdot d(\mathbf{w}_{k'}, \mathbf{x}_i)\right)}_{\text{"old" loss}} \tag{21}$$

Using the Lemma B.1 and Equation 21, we can observe that without the stop-gradient operator, underlying objective combining empirical (Monte-Carlo) estimate of conditional entropy of label given extracted features and "weighted" distance (*i.e.,* "old" loss).

**Discussion.** As noted by Boudiaf *et al.* [2], optimizing the conditional entropy loss requires special scrutiny, since the optima could result in trivial solutions on the simplex vertices *i.e.,* assigning all samples to a single class. Particularly, a small value of learning rate and fine-tuning the whole network (similar to [10]) are crucial to prevent dramatical deterioration in performance. In their experiments [2], they found that training the classifier with conditional entropy and cross-entropy loss signficant decrease the performance of few-shot learner. Beside, utilizing the conditional entropy for "push" loss does not affect the performance of classifier since it does not lead to collapsed solutions.

# C  Pseudo Code

We provide the pseudo-code of `POODLE` in a coding style similar to Pytorch as in Algorithm 1.

---

**Algorithm 1:** PyTorch-style pseudocode for `POODLE`.

```python
# f: classifier
# support: support images
# support_labels: labels of support images
# query: query images
# alpha: weight of positive term
# beta: weight of negative term

# sample negative data
neg_data = sample_from_train_set()

# construct positive data
if transductive:
    n_support = support.size(0)
    pos_data = concatenate([support, query])
else:
    pos_data = support

for i in range(n_steps):
    # compute logits
    pos_out = f(pos_data)
    neg_out = f(neg_data)

    # compute POODLE loss
    pull_loss = sum(pos_out * softmax(pos_out).detach())
    push_loss = sum(neg_out * softmax(neg_out).detach())
    poodle_loss = alpha * pull_loss - beta * push_loss

    # compute CE loss
    ce_loss = sum(support_label * log_softmax(pos_out[: n_support]))
    loss = ce_loss + poodle_loss

    # optimization step
    loss.backward()
    optimizer.step()
```

---

# D  Additional results

In this section, we provide more experimental results when applying `POODLE` for various network architectures and conducting ablation study to verify its effectiveness under numerous configurations. **Note that by `POODLE`, we refer to `POODLE`-B *i.e.*, negative samples are drawn from base classes, unless otherwise state.**

## D.1  Dataset

In this section, we briefly describe the datasets used for evaluating performance of *cross-domain* FSL below.

- iNatural-2017 [60]: This heavy-tailed dataset consists of $859,000$ images from over $5,000$ species of plants and animals. For *disjoint* negative sampling, we follow the meta-iNat benchmark [60, 58] *i.e.,* splitting the dataset to 908 classes for sampling negative samples and 227 classes for evaluation.

- EuroSAT [22]: This dataset covers total $27,000$ labeled images of Sentinel-2 satellite images, which consist of 10 classes and the patches measure $64 \times 64$ pixels.

- ISIC-2018 (ISIC) [7]: This dataset covers dermoscopic images of skin lesions. Precisely, we use the training set for task 3 (*i.e.,* lesion disease classification), which contains $10,015$ images with 7 ground truth classification labels.

It is worth mentioning that, for the sake of simplicity, we apply the same image transformation as in *mini*-Imagenet to these cross-domain datasets. Thus, the resolution of processed images of cross-domain datasets are $84 \times 84$ in our implementation. In contrast, the benchmark protocol using the image resolution of $224 \times 224$ [21]. A higher image resolution helps improve the performance of classifier significantly as we demonstrate in Section D.5.4. Nevertheless, POODLE successfully boosts the performance of all baselines in *cross-domain* FSL by a large margin.

## D.2 Standard FSL

In this section, we provide the experimental results of backbones other than Resnet-12. Specifically, we consider widely adopted architectures, namely Conv-4-512 [56], WideResnet [63], Mobilenet [25], and Densenet [27]. Particularly:

- **Conv-4-512**: We follow [56] to implement this architecture. More concretely, it is consisted of $4$ convolutional layers with hidden channels of $64$ and the output dimension is $512$.
- **WRN-28-10** [63]: We follow [49, 58] and use the wide residual network with $28$ convolutional layers and widening factor of $10$.
- **DenseNet-121** [27]: Similar to [58], we adopt the 121-layers architecture while removing the first two-down sampling layers and using the kernel size of $3 \times 3$ for the first convolutional layer.
- **MobileNet** [25]: We follow [58] and use the standard MobileNet for ImageNet [25] but remove the first two down-sampling layers of the network.

For all aforementioned architectures, we adopt the same training configuration of Resnet-12 as described in Section 5, excepts for WRN-28-10 we use the batch size of $32$ for all datasets. Beside that, we use the negative samples from base classes *i.e.,* POODLE-B in all experiments unless otherwise stated.

Table 7 shows the improvement of POODLE with various backbones of different network architectures on *mini*-Imagenet. We can see that our approach consistently enhance the accuracy of **all** baselines by a large margin *i.e.,* $2 - 4\%$ in 1-shot protocol and $0.5 - 1\%$ in 5-shot protocol.

In Table 8, we present the comparison between performance achieved by **WRN-28-10** with our approach and other algorithms on *mini-* and *tiered-*Imagenet. It can be observed that combining POODLE with other techniques such as SSL and KD achieves comparable or outperform state-of-the-art techniques for both inductive/transductive inference on two datasets.

## D.3 Results of reimplemented transductive algorithms

In this section, we report the accuracy of our reimplemented of transductive algorithms and their originally reported performance (balanced query set). In Table 9, we present the performance of our reproduced transductive algorithms (on Resnet-12). We can see that our implementation achieve comparable or higher accuracy than the original work even though we adopt the more efficient architecture (*i.e.,* Resnet-12 compared to Densenet).

## D.4 Imbalanced query set

So far, prior works in (transductive) FSL mainly concern with balanced query samples *i.e.,* numbers of query images for each class are equal. However, in practice, there is no guarantee that this setting is hold. In this section, we conduct experiments to benchmark the performance of different transductive algorithms under class-skew *i.e.,* imbalanced query data. We simulate the long-tail distribution of the query samples through the Dirichlet distribution. Precisely, for each novel task, we sample the number of queries for every category from a Dirichlet distribution with concentration $\kappa$ for all classes so that the total number of query images is always $75$ (similar to the experiments in the previous section).

In the "pull" loss term, we employ a *"soft"* weights for minimizing distance between prototypes and samples without any explicit regularization for distance between prototypes. In conventional setup for transductive learning, the number of query samples of each category are uniformly distributed, hence, the positive samples of each category implicitly constrains the prototype by "pulling" corresponding prototype. Under an extremely imbalanced query set, the positive samples from the dominated class

Table 7: Comparison to different baselines for the standard FSL (a.k.a in-domain FSL) on *mini-ImageNet* with various network architectures. The results are evaluated with $10,000$ random sampled tasks. The $95\%$ confidence interval of 1-shot and 5-shot classification are roughly $0.2$ and $0.1$, respectively.

| Network | Baseline | Variant | Inductive | | Transductive | |
|---|---|---|---|---|---|---|
| | | | 1-shot | 5-shot | 1-shot | 5-shot |
| WRN-28-10 | Simple | CE Loss | 61.23 | 81.08 | 61.23 | 81.08 |
| | | POODLE | **64.94** | **81.88** | **71.49** | **83.48** |
| | Rot | CE Loss | 64.77 | 83.66 | 64.77 | 83.66 |
| | | POODLE | **68.27** | **84.45** | **75.24** | **85.95** |
| | Rot + KD | CE Loss | 67.12 | 84.15 | 67.12 | 84.15 |
| | | POODLE | **69.67** | **84.84** | **77.30** | **86.34** |
| DenseNet-121 | Simple | CE Loss | 64.88 | 81.99 | 64.88 | 81.99 |
| | | POODLE | **66.53** | **82.55** | **75.55** | **84.62** |
| | Rot | CE Loss | 66.51 | 83.92 | 66.51 | 83.92 |
| | | POODLE | **68.68** | **84.52** | **77.85** | **86.49** |
| | Rot + KD | CE Loss | 68.66 | 84.17 | 68.66 | 84.17 |
| | | POODLE | **70.29** | **84.80** | **78.57** | **84.49** |
| Conv-4-512 | Simple | CE Loss | 50.90 | 70.44 | 50.90 | 70.44 |
| | | POODLE | **53.60** | **71.08** | **58.61** | **72.78** |
| | Rot | CE Loss | 51.36 | 70.81 | 51.36 | 70.81 |
| | | POODLE | **54.04** | **71.69** | **58.84** | **73.24** |
| | Rot + KD | CE Loss | 51.46 | 70.42 | 51.46 | 70.42 |
| | | POODLE | **53.87** | **71.36** | **58.50** | **72.87** |
| MobileNet | Simple | CE Loss | 60.99 | 77.45 | 60.99 | 77.45 |
| | | POODLE | **61.75** | **77.87** | **70.63** | **80.04** |
| | Rot | CE Loss | 64.57 | 80.86 | 64.57 | 80.86 |
| | | POODLE | **65.45** | **81.37** | **74.81** | **83.50** |
| | Rot + KD | CE Loss | 65.41 | 80.60 | 65.41 | 80.60 |
| | | POODLE-I | **66.02** | **81.18** | **74.55** | **83.09** |

might pull all the prototypes to their region by learning similar features of samples from different classes. In that case, increasing the value of $\beta$ to prevent the collapsed solution is necessary.

Table 10 presents the results of different transductive learning algorithms under the imbalanced query set. The detailed performance of the reproduced algorithms compared to the original papers are reported in Section D.3. The coefficient of "push" term is set to $\alpha = 1$. Since TIM [2] explicitly use the uniform distribution information of samples in the query set, its performance drastically drops in the imbalanced FSL. Other variants of K-means are more robust to the long-tail distribution, but they are far inferior to POODLE with $\beta = 0.75$.

## D.5 Ablation study

### D.5.1 Removing stop-gradient operator

As discussed before in Section B, the stop-gradient operator is of paramount important to avoid trivial solutions. In this section, we consider different choice of usage of the stop-gradient operator namely not using it in both "push/pull" terms, use only with one of two term, and use it for both terms in Table 11.

From the table, we can see that the stop-gradient operator does not affect the performance much in inductive inference. It can be explained as the posterior distribution of support data already has a high probability for "correct" classes and the conditional entropy term does not affect much. In

Table 8: Comparison to the state-of-the-art methods on *mini*-ImageNet, and *tiered*-Imagenet using **inductive** and **transductive** settings on **WRN-28-10**. The results obtained by our models (pear-shaded) are averaged over $10,000$ episodes.

| Method | Transd. | *mini*-ImageNet | | *tiered*-ImageNet | |
|---|---|---|---|---|---|
| | | 1-shot | 5-shot | 1-shot | 5-shot |
| LEO [49] | | 61.8 | 77.6 | 66.3 | 81.4 |
| SimpleShot [58] | | 63.5 | 80.3 | 69.8 | 85.3 |
| MatchNet [56] | ✗ | 64.0 | 76.3 | - | - |
| CC+rot+unlabeled [18] | | 64.0 | 80.7 | 70.5 | 85.0 |
| FEAT [62] | | 65.1 | 81.1 | 70.4 | 84.4 |
| Simple | | 61.23 | 81.08 | 67.63 | 83.93 |
| Simple + POODLE | | 64.94 | 81.88 | 70.25 | 84.64 |
| Rot + KD | | 67.12 | 84.15 | - | - |
| Rot + KD + POODLE | | **69.67** | **84.84** | - | - |
| AWGIM [20] | | 63.1 | 78.4 | 67.7 | 82.8 |
| Ent-min [10] | | 65.7 | 78.4 | 73.3 | 85.5 |
| SIB [26] | | 70.0 | 79.2 | - | - |
| BD-CSPN [38] | | 70.3 | 81.9 | 78.7 | 86.92 |
| LaplacianShot [66] | ✓ | 74.9 | 84.1 | 80.2 | 87.6 |
| TIM-ADM [2] | | 77.5 | 87.2 | 82.0 | 89.7 |
| TIM-GD [2] | | **77.8** | **87.4** | **82.1** | **89.8** |
| Simple + POODLE | | 71.49 | 83.48 | 76.27 | 85.83 |
| Rot + KD + POODLE | | 77.30 | 86.34 | - | - |

Table 9: Results of our implementations of various **transductive** methods compared to original works on *mini*-ImageNet. The results of our implementations (pear-shaded) are evaluated on *Resnet-12 +* Rot + KD baseline with 2,000 random sampled tasks.

| Methods | Network | 1-shot | 5-shot |
|---|---|---|---|
| Mean-shift | Resnet-12 | $73.49 \pm 0.55$ | $84.24 \pm 0.32$ |
| Mean-shift [36] | DenseNet-121 | $71.39 \pm 0.27$ | $82.67 \pm 0.15$ |
| Bayes k-means | Resnet-12 | $71.40 \pm 0.50$ | $83.79 \pm 0.31$ |
| Bayes k-means [36] | DenseNet-121 | $72.05 \pm 0.24$ | $80.34 \pm 0.17$ |
| Soft k-means | Resnet-12 | $74.55 \pm 0.49$ | $84.53 \pm 0.30$ |
| Soft k-means [46] | Conv-4-64 | $50.09 \pm 0.45$ | $64.59 \pm 0.28$ |
| CAN_T | Resnet-12 | $71.04 \pm 0.53$ | $84.21 \pm 0.30$ |
| CAN_T [24] | Resnet-12 | $67.19 \pm 0.55$ | $80.64 \pm 0.35$ |
| TIM | Resnet-12 | $77.69 \pm 0.55$ | $87.40 \pm 0.29$ |
| TIM [2] | Resnet-18 | $73.90 \pm n/a$ | $85.00 \pm n/a$ |

transductive inference, removing the stop-gradient operator on "pull" term results in noticeable decrease in accuracy as we articulate in Section B. On the other hand, taking out stop-gradient operator on "push" term does not worsen the performance since it does not lead to trivial solutions.

### D.5.2 Sensitive to $\alpha$ and $\beta$

So far, we only fixed the coefficient of "push/pull" loss with $\alpha = 1$ and $\beta = 0.5$, we now consider different configurations of these parameters to understand the sensitive of POODLE to these hyper-parameters. Precisely, we report the accuracy gain of POODLE with different combination of $\alpha$ and $\beta$ taken from the list $[0.0, 0.25, 0.5, 0.75, 1.0, 2.0, 5.0]$ as in Figure 3.

We can observe from Figure 3 that POODLE is robust to the change of $\alpha$ and $\beta$: the accuracy of network is improved as long as the value of $\alpha$ is larger than $\beta$. Furthermore, the improvement in accuracy usually does not change much when varying $\alpha$ and $\beta$.

Table 10: Results of transductive learning on imbalance **query** data of *mini*-ImageNet. The results obtained by our models (blue pearl-shaded) are averaged over 2,000 episodes. $\kappa$ denotes the concentration parameter of the Dirichlet distribution.

| | $\kappa = 0.5$ | | $\kappa = 1$ | | $\kappa = 2$ | | $\kappa = 5$ | |
| Methods | 1-shot | 5-shot | 1-shot | 5-shot | 1-shot | 5-shot. | 1-shot | 5-shot |
|---|---|---|---|---|---|---|---|---|
| Inductive | 66.91 | 82.75 | 66.16 | 82.62 | 66.10 | 83.01 | 66.23 | 83.16 |
| Mean-shift [36] | 68.88 | 80.54 | 70.31 | 81.82 | 71.78 | 83.16 | 72.92 | 83.98 |
| Bayes k-means [36] | 69.69 | 82.86 | 69.91 | 82.95 | 70.64 | 83.54 | 71.39 | 83.83 |
| Soft k-means [46] | 65.31 | 75.65 | 68.61 | 78.90 | 71.56 | 81.58 | 73.45 | 83.33 |
| CAN_T [24] | 72.30 | 83.53 | 71.28 | 83.62 | 71.09 | 84.06 | 71.18 | 84.25 |
| TIM [2] | 47.65 | 52.79 | 54.86 | 61.32 | 61.84 | 68.73 | 68.74 | 76.39 |
| POODLE-B $\beta = 0.00$ | 61.16 | 76.40 | 66.02 | 79.88 | 70.03 | 82.43 | 73.03 | 84.23 |
| POODLE-B $\beta = 0.50$ | 64.56 | 81.01 | 69.14 | 83.63 | 72.91 | 85.14 | **75.79** | 85.78 |
| POODLE-B $\beta = 0.75$ | 75.20 | 86.51 | **74.06** | 86.17 | **74.77** | **86.48** | 74.74 | **86.34** |
| POODLE-B $\beta = 1.00$ | **75.48** | **88.36** | 71.14 | **86.31** | 70.20 | 85.53 | 68.94 | 84.31 |

Table 11: Results of different settings of stop-gradient in push and pull terms. The results are evaluated with $2,000$ random sampled tasks on *Resnet-12* with Rot + KD baseline. The $95\%$ confidence interval of 1-shot and 5-shot classification are roughly 0.45 and 0.3 for **inductive** setting and 0.6 and 0.4 for **transductive**, respectively. Results of inductive baseline for 1-shot and 5-shot are $65.91 \pm 0.44$ and $82.95 \pm 0.30$, respectively.

| | **Stop-grad** | | **Inductive** | | **Transductive** | |
| Method | Pull | Push | 1-shot | 5-shot | 1-shot | 5-shot |
|---|---|---|---|---|---|---|
| POODLE-B | ✓ | ✓ | 67.52 | 83.71 | 77.58 | 85.87 |
| | ✗ | ✓ | 67.50 | 83.56 | 62.04 | 82.87 |
| | ✓ | ✗ | 67.37 | 83.73 | 78.51 | 86.28 |
| | ✗ | ✗ | 67.37 | 83.64 | 72.04 | 85.08 |
| POODLE-R | ✓ | ✓ | 67.80 | 83.50 | 77.25 | 85.52 |
| | ✗ | ✓ | 67.80 | 83.39 | 66.55 | 83.65 |
| | ✓ | ✗ | 67.77 | 83.43 | 78.11 | 85.83 |
| | ✗ | ✗ | 67.77 | 83.36 | 73.29 | 84.77 |

### D.5.3 Number of sampled negative data

Intuitively, a larger number of negative samples will provide more informative cues to the classifer, however, they might require a notable memory footprint and have significant latency. We conduct experiment to quantify the impact of number of negative samples to the accuracy of POODLE in Table 12. These results indicate that higher number of negative samples consistently leads to better performance, however, the gain in accuracy is relatively small.

Table 12: Results of our methods with different number of out-of-distribution samples per task. The results are evaluated with $10,000$ random sampled tasks on *Resnet-12* with Rot + KD baseline. The $95\%$ confidence interval of 1-shot and 5-shot classification are roughly 0.20 and 0.10 for both inductive and transductive inference. Results of inductive baseline for 1-shot and 5-shot are $66.32 \pm 0.20$ and $82.99 \pm 0.13$, respectively.

| Number of | **1-shot** | | **5-shot** | |
| negative samples | POODLE-I | POODLE-T | POODLE-I | POODLE-T |
|---|---|---|---|---|
| 50 | 67.61 | 77.38 | 83.54 | 85.56 |
| 100 | 67.71 | 77.61 | 83.62 | 85.64 |
| 200 | 67.77 | 77.69 | 83.69 | 85.70 |
| 500 | **67.84** | **77.73** | **83.71** | **85.73** |

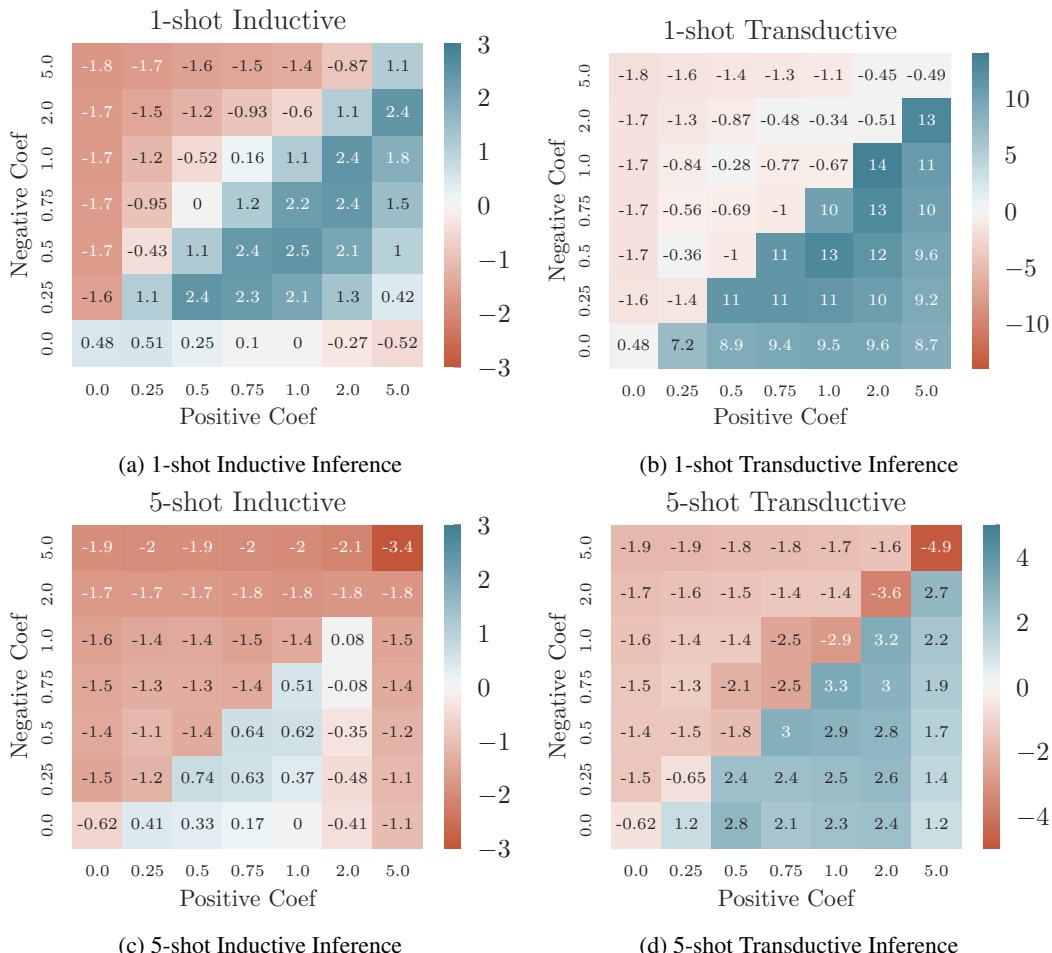

Figure 3: Ablation study on the affect of positive and negative coefficient of POODLE (*i.e.,* $\alpha$ and $\beta$) to **accuracy gain** of (simple baseline) **Resnet-12** on *mini*-Imagenet compared to no fine-tuning at all with $10,000$ random tasks. The baseline where we directly assign the prototype of each class to the mean of its support images obtains accuracy of $61.63 \pm 0.20$ and $80.78 \pm 0.11$ for 1-shot and 5-shot classification, respectively.

### D.5.4   Higher image resolution

Existed works in FSL usually employ the standard image pre-processing schemes such as resizing to $84 \times 84$ pixels. We consider using a slightly different setup where we resize the image to a higher resolution than $84$ during training and testing. We keep all other hyper-parameters and configurations as in *simple baseline*. We report the accuracy of these models with POODLE in both inductive and transductive inference in Table 13. It can be observed that the higher resolution usually leads to better performance. Nevertheless, POODLE successfully raises the accuracy in all cases up to $5\%$ in inductive 1-shot classification.

### D.5.5   Different number of finetuning steps

In previous experiments, we fixed the number of fine-tuning steps when employing POODLE with $T = 250$ steps. We now consider different numbers of fine-tuning steps and evaluate its impact on the final performance of the classifier. Particularly, we report the accuracy of the classifier fine-tuned with POODLE and standard cross-entropy loss with various number of steps in Table 14. In our experiments, we find that POODLE is not sensitive to number of gradient updates and do not requires a high number of steps to achieves good performance. This finding allows us to accelerate the fine-tuning steps and reduce the computational cost for inference phase.

Table 13: Comparing the performance of simple baseline on *mini*-Imagenet with different image resolutions. POODLE-I and POODLE-T indicate the result of our algorithm in **inductive** and **transductive** inference. The results are evaluated with $10,000$ random sampled tasks. The $95\%$ confidence interval of 1-shot and 5-shot classification are roughly $0.2$ and $0.1$, respectively. **Bold** numbers indicate significant improvement (*i.e.,* p-value $\leq 0.05$) compared to *weakest* corresponding entries.

| Network | Resolution | 1-shot | | | 5-shot | | |
| | | CE Loss | POODLE-I | POODLE-T | CE Loss | POODLE-I | POODLE-T |
|---|---|---|---|---|---|---|---|
| Resnet-12 | $84 \times 84$ | 61.63 | 64.08 | 74.46 | 80.78 | 81.41 | 83.70 |
| | $140 \times 140$ | 62.66 | 65.90 | 75.08 | 82.06 | 82.89 | 84.88 |
| | $180 \times 180$ | **64.85** | 67.05 | **76.17** | 82.52 | 83.19 | **85.31** |
| | $224 \times 224$ | 62.90 | **67.22** | 74.46 | **82.59** | **83.61** | 85.23 |
| WRN-28-10 | $84 \times 84$ | 61.23 | 64.94 | **71.49** | 81.08 | 81.88 | 83.48 |
| | $140 \times 140$ | **61.76** | 66.27 | 71.23 | **81.73** | **82.70** | **84.05** |
| | $180 \times 180$ | 61.72 | **66.34** | 70.61 | 81.67 | 82.65 | 83.90 |

Table 14: Results of our methods with different number of update iterations. The results are evaluated with $2,000$ random sampled tasks on *Resnet-12* with Rot + KD baseline. The $95\%$ confidence interval of 1-shot and 5-shot classification are roughly $0.45$ and $0.3$ for **inductive** setting and $0.5$ and $0.3$ for **transductive**, respectively. Results of inductive baseline for 1-shot and 5-shot are $65.91 \pm 0.44$ and $82.95 \pm 0.30$, respectively. **Bold** numbers indicate significant improvement (*i.e.,* p-value $\leq 0.05$) compared to *weakest* corresponding entries.

| Finetuning steps | 1-shot | | | 5-shot | | |
| | CE Loss | POODLE-I | POODLE-T | CE Loss | POODLE-I | POODLE-T |
|---|---|---|---|---|---|---|
| 50 | 66.18 | 67.61 | 74.75 | **83.05** | 83.72 | 85.57 |
| 100 | 66.17 | 67.54 | 77.03 | 82.94 | 83.71 | 85.78 |
| 250 | 66.17 | 67.52 | 77.58 | 82.76 | 83.69 | 85.87 |
| 400 | 66.17 | 67.49 | **77.63** | 82.68 | 83.70 | 85.87 |