# OpenReview forum: "POODLE: Improving Few-shot Learning via Penalizing Out-of-Distribution Samples"
_NeurIPS.cc/2021/Conference — NeurIPS 2021 Poster_

### Official Review · Reviewer_FGzB · 2021-07-12

**Rating:** 4
**Confidence:** 4

**Summary:**

In this paper, authors have leveraged out-of-distribution samples to improve the generalization of few-shot learners. The main idea is to use the large margin principle in metric learning and add terms to the loss function to encourage the learned prototypes to be far from the out-of-distribution samples and to be close to support data. In summary, in addition to standard cross-entropy loss, authors have used to extra loss functions for penalizing  the large distance between the embeddings of two nearby neighbors, and the small distance between the embeddings of samples of difference classes.


**Limitations And Societal Impact:**

Authors have not talked about potential societal impacts such as fairness of their method. It would be great if a short discussion is added on the limitations of the proposed approach.

**Main Review:**

The paper is well-written and presents an interesting idea and direction. However, there are few issues that make the paper not ready for publication in NeurIPS.

- The core idea of using large-margin principle is not novel and as authors have mentioned, it was first introduced in "large margin nearest neighbors" papers.
- The regularization terms (i.e., the push and pull loss functions) have introduced to new hyper-parameters (\alpha and \beta) but there is no discussion on how these two could be chosen effectively.
- In Figure 2, authors have mentioned that their method "consistently yields accuracy gain on different backbones" compared to other approaches. However, there is some issue with the MobileNet case.
- Comparing to other SOTA methods, authors have mentioned: "We can see that our approach remarkably improves the performance of the baseline and achieves a comparable performance with the state-of-the-art approaches in the tiered-ImageNet." However, this is not true. the difference between other SOTA methods and the proposed approach on the tiered-Imagenet is basically the same as the difference between other approaches and SOTA methods on mini-Imagenet and CUB. Therefore, the performance of the proposed approach is on par with other SOTA methods, roughly speaking.
- Re table 5, authors have mentioned if we increase the coefficient of the push, we discourage the prototypes from learning the aforesaid features and this will help the POODLE method to gain better performance. However, in most of the cases \beta=0.75 is giving better results compared to \beta=1 and this is not explained.
---------------------------------
Post-Rebuttal:
- Thanks to authors for their comments regarding my concerns. I still believe that the novelty of this paper is limited and the presentation has issues which make the paper not ready for NeurIPS. I highly recommend that authors address the concerns that reviewers brought up and submit to the next ML venue.


**Time Spent Reviewing:**

5

---

> ### Author Response · Authors · 2021-08-10
> **Thanks for your detailed review.**
>
> ### 1. The core idea of using large-margin principle is not novel and as authors have mentioned, it was first introduced in "large margin nearest neighbors" papers.
>
> Please see our response to common issues.
>
> ### 2. The regularization terms (i.e., the push and pull loss functions) have introduced to new hyper-parameters ($\alpha$ and $\beta$) but there is no discussion on how these two could be chosen effectively.
>
> We will provide additional details on hyper-parameters tuning in the revised paper. Particularly, we use a validation set (16 novel classes that are distinct from both training and testing classes in the case of mini-Imagenet) to tune these values similar to convention practice in transductive learning such as [1, 2]. In our experiments, we only tune $\alpha=1$ and $\beta=0.5$ for (simple) Resnet-12 on the mini-Imagenet dataset and use these values across all settings (in contrast to some work that involves a different set of hyperparameters each experiment e.g, FEAT). We also provided the ablation study for different values of $\alpha$ and $\beta$ in section D.4.2 of the supplementary document.
>
> ### 3. Comparing to other SOTA methods, authors have mentioned: "We can see that our approach remarkably improves the performance of the baseline and achieves a comparable performance with the state-of-the-art approaches in the tiered-ImageNet." However, this is not true. the difference between other SOTA methods and the proposed approach on the tiered-Imagenet is basically the same as the difference between other approaches and SOTA methods on mini-Imagenet and CUB. Therefore, the performance of the proposed approach is on par with other SOTA methods, roughly speaking.
>
> Although it is true that our results are only comparable with SOTA on tiered-Imagenet as the reviewer has pointed out, we indeed achieve higher performance than SOTA on mini-Imagenet and CUB as can be seen in our response to common issues (rot + POODLE Resnet-12) and Table 3 in the paper. Also, the combination of KD + SSL + POODLE, which has a good margin over other approaches, can also be considered as our (minor) contribution.
>
> *That being said, we would tone down the statement in the revised paper.* Furthermore, our work is orthogonal to most work in the literature of FSL, thus, the final performance should not be compared directly like that.  Instead, we argue that the additional gain when leveraging our method (on top of off-the-shelf approaches) is more desirable. In that aspect, the response to common issues, Table 1 (and Table 6 in supplementary) have indicated the effectiveness of POODLE.
>
>
> ### 4. Table 5, authors have mentioned if we increase the coefficient of the push, we discourage the prototypes from learning the aforesaid features and this will help the POODLE method to gain better performance. However, in most of the cases, $\beta=0.75$ is giving better results compared to $\beta=1$ and this is not explained.
>
> Although increasing $\beta$ can help in the imbalance setup, having too high value for the push term encourages the prototypes to focus on maximizing distances to all OOD samples. Thus, the classifier can learn an arbitrary representation as long as it is far from sampled OOD samples and the learned representation does not necessarily be close to the positive data. In summary, we need to balance these two objectives.
>
> **References**
>
>    [1] LaplacianShot: Laplacian Regularized Few Shot Learning (ICML 2020)
>
>    [2] Transductive Information Maximization For Few-Shot Learning (NeurIPS 2020)

---

### Official Review · Reviewer_FMo7 · 2021-07-16

**Rating:** 5
**Confidence:** 4

**Summary:**

The proposed approach leverage the out-of-distribution (OOD) samples to improve the few-shot learning (FSL). The OOD samples were exploited from easily available base classes, although random uniform sampling shows better results. Maximizing the distance between the OOD and positive samples helps to mitigate the ambiguity between the classes. The proposed approach is applicable for the inductive and transductive settings.

**Main Review:**

The paper has the following pros and cons:
Pros:
1: The proposed model is simple. Negative samples are easy to obtain.
2: The paper is written well and easy to understand.
3: Good result in both (inductive and transductive) settings.

Cons:
1: The paper's main contribution is to use the negative samples to design a margin-based loss that minimizes the confusion between similar contexts. The margin-based loss is not novel. Many previous supervised learning approaches use this type of loss frequently. In the FSL setting, it may be novel, but this contribution is not significant.

2: The result in Table-3 and Table-4 are confusing. Paper reported result by a combination of Rot+KD, which is not a fair comparison with the previous approach. The earlier method can be significantly improved using the self-supervised learning model. The paper combines the earlier model self-supervision, KD and max-margin loss to improve the result. There is no significant improvement in terms of methodology for the FSL. The results reported in the Table-1 and 2 with

3: Few popular are ignored and not compared in the paper, e.g. [1] [2] [3]. I request to the author please compared these approaches and also include the recent baseline.


[1] Adaptive Subspaces for Few-Shot Learning, CVPR-20
[2] META-LEARNING WITH LATENT EMBEDDING OPTIMIZATION, ICLR-19
[3] Attentive Weights Generation for Few Shot Learning via Information Maximization, CVPR-20
[4] ECKPN: Explicit Class Knowledge Propagation Network for Transductive Few-shot Learning, CVPR-21




**Time Spent Reviewing:**

4-5

---

> ### Author Response · Authors · 2021-08-10
> **Thanks for your constructive review**
>
> ### 1. The margin-based loss is not novel. Many previous supervised learning approaches use this type of loss frequently
>
> Please see our response for the common issues.
>
> ### 2. The result in Table-3 and Table-4 are confusing. Paper reported result by a combination of Rot+KD, which is not a fair comparison with the previous approach. The earlier method can be significantly improved using the self-supervised learning model. The paper combines the earlier model self-supervision, KD and max-margin loss to improve the result.
>
> Please see our response for the common issues. In addition, we also conduct experiments to compare our methods with some related work in response to the first question of Reviewer 5hh7.
>
> Also, note that all methods in Table 4 use SSL with 20% unlabeled data from the same domain. Hence, the comparison between POODLE and prior work is fair. Furthermore, our main purpose is to demonstrate the significant improvement from POODLE on the baseline in Cross-domain FSL with a very small additional cost during inference (see our response to question 5 of reviewer 5hh7).
>
> ### 3. Few popular are ignored and not compared in the paper, e.g. [1] [2] [3]. I request to the author please compared these approaches and also include the recent baseline
>
> We have added those baselines along with much recent work in the response to the common issues. We report for ECKPN and transductive version of DSN in the below table. Similar to Table 3, we report the transductive POODLE-B with a **simple** baseline for a fair comparison with other methods.
>
> TABLE 1: Transductive **mini-Imagenet** with **Resnet-12** backbone. Our results are evaluated with **10,000** episodes with the same configuration as specified in the paper (see section 5.1).
>
> | Method            | 1-shot          | 5-shot          |
> | ----------------- | --------------- | --------------- |
> | ECKPN [4]         | 70.48±0.38     | **85.42±0.46** |
> | DSN-MR [1]        | 64.60±0.72     | 79.51±0.50     |
> | simple + POODLE-B | **74.47±0.24** | 83.80±0.13     |
>
> TABLE 2: Transductive **tiered-Imagenet** with **Resnet-12** backbone. Our results are evaluated with **10,000** episodes with the same configuration as specified in the paper (see section 5.1).
>
> | Method            | 1-shot          | 5-shot          |
> | ----------------- | --------------- | --------------- |
> | ECKPN [4]         | 73.59±0.45     | **88.13±0.28** |
> | DSN-MR [1]        | 67.39±0.82     | 82.85±0.56     |
> | simple + POODLE-B | **78.57±0.24** | 86.61±0.15     |
>
> **References**
>
>    [1] Adaptive Subspaces for Few-Shot Learning (CVPR 2020)
>
>    [4] ECKPN: Explicit Class Knowledge Propagation Network for Transductive Few-shot Learning (CVPR 2021)

---

> > ### Comment · Reviewer_FMo7 · 2021-08-18
> > **Re: Rebuttal Response**
> >
> > Thanks for the experiment and clarification.
> > I am convinced with the fair comparison, still, have concerns about the novelty/methodology.
> > The max-margin loss helps to detect the outlier but still compared to standard neural network it does show high confidence for the outlier sample i.e. softmax shows high confidence towards some training classes.  Like the Bayesian model, it does not have an advantage that for the outlier entropy is very high.
> >
> > Overall, empirically results looks good but novelty and contribution is still a concern.
> > I will increase my score from reject to weak reject.
> >
> > Thanks

---

### Official Review · Reviewer_C2g5 · 2021-07-16

**Rating:** 6
**Confidence:** 3

**Summary:**

The paper considers the problem of few-shot learning, and proposes to leverage out-of-distribution data as negative examples to regularize the few-shot model. The intuition is that in cases where there is ambiguity because of lack of training samples, the out-of-distribution data as counter examples, would help the model eliminate the less generalizable solutions.

The out-of-distribution data here refers to samples from classes other than classes of the target task.There is an emphasis that these counter examples should be from the same domain as the target task.

The few-shot learning setup in this paper is as follows:
1. Use a frozen pre-trained model to get the representation of given data points from the target task.
2. Learn a prototype representation for each class, `W = {w1,...wk}` (this is initialized to the mean of the representations of the examples from the training set for each class).
3. Given a query example, compute the distance of its representation with the prototype representation of each class, and apply softmax on the negative scaled distance scores to get predictive probability distribution over all classes.

In the standard case, the few-shot classifier (with W, and a scalar that is applied to the distance scores as parameters) is simply trained with cross entropy loss. This paper suggests adding an auxiliary loss inspired by Large Margin Nearest Neighbors (Equation 3 and 4 in the paper).

The paper considers both in-domain and cross domain FSL. For in-domain FSL, they sample negative examples from pre-training data. For cross-domain FSL, since pre-training data has a different domain than the target task, they propose to use a set of samples drawn from classes that are disjoint to the target task. In cases where such negative examples might not be available they propose using a set of unlabeled data of a set of classes from the target domain that overlap with target classes.

Additionally, the authors discuss the case where there is class imbalance in the few-shot training samples. They propose having higher weights for the push terms of the regularizer loss to avoid the prototype representations to collapse.



**Limitations And Societal Impact:**

The authors discuss the potential problem if the `number of few shot samples is not balanced` over classes. What happens in this case is that all prototypes are learned to have high similarity to the samples of dominated classes and the predicted classes. Hence, the prototype features would be similar across classes. The authors argue that in this case, increasing the coefficient of the push term of the loss helps.
Another point that is implicitly mentioned is the `sensitivity of the proposed technique the negative sample selection strategy`.
Finally, I think I would add `lack of theoretical support for the proposed method` to this list.

Societal impact: The authors have generally explained the importance of having few shot learning models.


**Main Review:**

**Originality:** The main idea in the paper is based on contrasting between representations of in and out of distribution samples from the target domain (where the out of distribution-ness is rooted in the samples belonging to a separate set of classes). Finding a way to realize this idea inspired by LMNN, and evaluating its effectiveness empirically is the main contribution of this paper.

**Quality and Clarity**: I find some of the arguments and claims  in the paper a bit vague.

In the result provided in Table 1 in the paper, POODLE-R performs comparatively with the POODLE-B. First of all I am not sure I correctly understand what happens in POODLE-R, does that mean you simply sample from a uniform d-dimensional distribution?
I think there should be a more elaborate explanation for this than hypothesizing that the normalized representations of the samples extracted from the base classes are uniformly distributed. If simply samples from a uniform distribution is enough to provide “negative examples” to the auxiliary loss function, how does this fit into the intuitions provided in the paper behind why this method would work to begin with? and why does this not work for all cases, e.g., result in Table 4?

In line 42 as one the contributions of the paper the authors mentions:
`We propose a novel yet simple approach to learn the inductive bias of deep neural networks for FSL by leveraging out-of-distribution data.`
I can not figure out how to interpret this sentence. As the inductive bias is not something that the model learns from data and is not clear to me what the inductive bias refers to here.

In line 167 the authors claim
`We note that removing SG[·] in Eq. 3 would result in a different underlying objective, which empirically leads to a decrease in performance.`
I understand how the stop gradient operation in  Eq. 3 could lead to a difference, but appreciate it if the authors could explain how removing it from the equation leads to a different objective?

**Significance** :The method seems to be applicable on top of other techniques to improve FSL performance, and the results show significant improvements over baselines based on the numbers reported in the paper.


**Time Spent Reviewing:**

4h

---

> ### Author Response · Authors · 2021-08-10
> **Thanks for your adept response**
>
> ### 1. POODLE-R: does that mean you simply sample from a uniform d-dimensional distribution?
>
> We uniformly sample the vector with the same dimension as the output of the last layer of the network to construct negative samples (since we apply l2-normalization to this vector, the range of uniform distribution is not important).
>
> ### 2. If simply samples from a uniform distribution is enough to provide “negative examples” to the auxiliary loss function, how does this fit into the intuitions provided in the paper behind why this method would work to begin with? and why does this not work for all cases, e.g., result in Table 4?
>
> **Why POODLE-R works**: Our use of uniform distribution as negative examples is inspired by that feature uniformity is a desirable property for contrastive loss [1, 2], and so a good representation prefers such uniformity. The problem of uniformly distributing points on the unit hypersphere can be formulated as minimizing pairwise loss wrt some kernel functions [3, 4], which links well to the theory of supervised classification with softmax and cross-entropy (equivalent to minimizing pairwise loss or maximizing mutual information) [5, 6].
>
> Therefore, using uniform distribution as negative samples works for in-domain FSL because they will approximate the features of samples from base training data. For cross-domain FSL, POODLE-R will fail because the random features (which are similar to samples from the training domain) have a large discrepancy with the target domain, thus, not inducing meaningful cues. *Intuitively, the more similar between domains of OOD and test samples, the higher performance gain POODLE can achieve.*
>
> **Behavior of POODLE with different domains of OOD samples**: The above argument can be further proven empirically by a simple experiment: we train our classifier and compare the performance when using random uniform distribution and other datasets as negative examples. Precisely, we evaluate **simple Resnet-12** trained on **mini-Imagenet** with 10,000 episodes in the 1-shot inductive protocol. The OOD/test samples are drawn from the standard train/test split of each dataset, respectively. The 95% confidence interval is roughly 0.20 for all results.
>
> - In-domain FSL: As can be seen, using uniform sampled examples has the best and second-best accuracy when tested on mini-ImageNet and tiered-ImageNet, respectively. We can see a consistent trend: the more similar between test and negative data (mini-Imagenet = tiered-Imagenet > CUB > EuroSAT), the more accuracy gain we obtain. In addition, it's expected that using negative samples from tiered-Imagenet has a lower performance gain compared to using samples from mini-Imagenet when testing on mini-Imagenet. This is because some negative samples of tiered-Imagenet could be misclassified and indeed, not OOD samples.
>
> |  Test dataset               | Baseline classifier without OOD |        w/ uniform random distribution as OOD         | w/ mini-Imagenet as OOD                | w/ tiered-Imagenet as OOD               | w/ CUB as OOD                           | w/ EuroSAT as OOD                     |
> | --------------- | :---------------: | :---------------------------: | ----------------------------- | ----------------------------- | ----------------------------- | ----------------------------- |
> | mini-Imagenet   |       61.63       |           **64.30**           | 64.08                         | 63.72                         | 62.71                         | 61.76                         |
> | tiered-Imagenet |       63.04       |             64.28             | 64.01                         | **64.50**                     | 63.85                         | 62.78                         |
>
>
> - Cross-domain FSL: using random uniform features does not work because the discrepancy between the source and target domain is extremely large, e.g. animal (mini-Imagenet) vs satellite images (EuroSAT). This is reflected in the following experiment where we train on mini-ImageNet but test on CUB and EuroSAT which has a different domain.
>
>
> |  Test dataset               | Baseline classifier without OOD |        w/ uniform random distribution as OOD         | w/ mini-Imagenet as OOD                | w/ tiered-Imagenet as OOD               | w/ CUB as OOD                           | w/ EuroSAT as OOD                     |
> | --------------- | :---------------: | :---------------------------: | ----------------------------- | ----------------------------- | ----------------------------- | ----------------------------- |
> | CUB             |       48.55       |             48.88             | 49.01                         | 49.10                         | **51.40**                     | 49.18                         |
> | EuroSAT         |       65.18       |             63.85             | 64.58                         | 64.25                         | 64.70                         | **66.04**                     |
>
>
>
>
>
>
> ### 3. I can not figure out how to interpret this sentence. As the inductive bias is not something that the model learns from data and is not clear to me what the inductive bias refers to here.
>
> In our opinion, “inductive bias” may have different connotations across different contexts e.g., the structural bias imposed by the architecture, the prior knowledge of the target tasks, preference of the training scheme (e.g. what to learn first in curriculum learning). In the paper, we have defined inductive bias as a strategy to prioritize a hypothesis over others. We agree it can be obscure and will amend that statement.
>
> ### 4. I understand how the stop gradient operation in Eq. 3 could lead to a difference, but appreciate it if the authors could explain how removing it from the equation leads to a different objective?
>
> We provided an explanation in Section B of supplementary material. As a quick summary, omitting the stop-gradient induces an additional term that penalizes entropy of the prediction. The entropy loss needs additional tuning/regularization (e.g. batch-norm, etc) to prevent catastrophic degradation in performance as already shown in [7].
>
> **References**
>
>    [1] Understanding Contrastive Representation Learning through Alignment and Uniformity on the Hypersphere (ICML 2020)
>
>    [2] Intriguing Properties of Contrastive Losses
>
>    [3] Discrete energy on rectifiable sets
>
>    [4] Foundations of modern potential theory
>
>    [5] Rethinking Softmax with Cross-Entropy: Neural Network Classifier as Mutual Information Estimator
>
>    [6] A unifying mutual information view of metric learning: cross-entropy vs. pairwise losses (ECCV 2020)
>
>    [7] Transductive Information Maximization For Few-Shot Learning (NeurIPS 2020)

---

> > ### Comment · Reviewer_C2g5 · 2021-08-27
> > **Thanks for the response.**
> >
> > Thank you for the elaborate response. I reread the paper, all the reviews and rebuttal responses. I appreciate the clarifications and the additional experiments.
> >
> > **About Why POODLE-R works:**
> > - Thanks for the clarification. I see that your story is perfectly aligned with the results you have provided in the rebuttal. But I am still a bit confused, what differentiates the uniformly sampled representation from the representations for the examples of the new classes?
> > The way that I understand it, this would be some sort of regularization, and so the initial intuition that the model benefits from contrasting positive and negative samples does not hold in this case anymore. But in that case I don't understand why the results are different for in domain and out of domain examples.
> >
> > **Novelty:** (since it is an issue raised by other reviewers)
> > - The main contribution of the paper being to show how large margin nearest neighbor can be applied in the few shot learning setup and that it works, if the experiments are extensive enough and convincing, I find the novelty of the paper sufficient for this venue.
> > - However, I did a quick search around the topic and came across [1], and I think this is not cited in the paper although seems very relevant . I appreciate if the authors can discuss the connection of their work with this paper.
> >
> > **Points for improvements:**
> > -To me it seems, it should be possible and not be costly, to conduct some experiments on the available state of the art pretrained models. It would be nice if you can show that this approach is still useful, in the scaled up setting.
> >
> > **Final thought:**
> > I'd be happy to increase my score if the concerns about the novelty of the idea are addressed.
> >
> > [1] [Large Margin Few-Shot Learning, Yang Liu et al. (2018)](https://arxiv.org/pdf/1807.02872.pdf)

---

> > > ### Author Response · Authors · 2021-08-29
> > > **Additional clarifications**
> > >
> > > We thank the reviewer for spending substantial time/effort to evaluate our work and give response, we add more clarifications for your concerns below
> > >
> > > ### 1. But I am still a bit confused, what differentiates the uniformly sampled representation from the representations for the examples of the new classes?
> > >
> > > Under the limited data, the prototype can capture features that are weakly correlated to labels, e.g., grass in the background, etc. Note that these features can be discriminative with scarce data e.g., classifying rabbits and cars but we only have a handful of car images on the highway, the grass in the background can be discriminative. These weakly correlated features can cause confusion because samples from different classes might share them. Thus, the prototypes can have poor decision boundaries.
> > >
> > > Intuitively, as the uniformly sampled representation approximates the base training data, these OOD samples would also model weakly correlated features (in high-dimensional space). The small distance between prototypes and OOD samples are usually caused by weakly correlated features, as the positive samples will be not likely to share strongly correlated feature with distractor samples. Thus, pushing the prototypes away from OOD samples while pulling them to their support samples could move the prototype to the region to favor samples of that class and avoid the boundary (where samples share weakly correlated features). Consequently, fine-tuning process generates better decision boundaries.
> > >
> > > ### 2. However, I did a quick search around the topic and came across [1], and I think this is not cited in the paper although seems very relevant . I appreciate if the authors can discuss the connection of their work with this paper.
> > >
> > > Thanks for your suggestion, we clarify the novelty of the paper hereafter:
> > >
> > > - We would like to clarify that studying the large margin principle is *not* the target of this paper. Specifically, the key contribution of our paper is the idea of capitalizing OOD samples to tackle the ambiguity of FSL (also see the first answer of response to common issues). We believe this novel concept is general and margin loss (as done in this paper) is just one approach to realize that idea.
> > > - Regarding the relevant paper the reviewer pointed out, we would like to articulate the difference:
> > >   - The aforementioned paper [1] utilizes margin loss in a similar manner to [2] i.e., learning the **feature extractor** $f_\theta(.)$ by maximizing the distance between the representation of samples from different (and known) classes. Thus, this can be seen as an auxiliary loss in the pre-training phase. In our experience, maximizing the distance between support images similar to this mechanism when fine-tuning doesn't bring improvement (also see our answer to question 1 of reviewer 5hh7).
> > >   - In contrast, our paper aims to *leverage OOD samples* to refine the **prototype** vector of each class in novel tasks (i.e., a post-processing method). Precisely, we enforce the prototypes to have a large distance to those distractor samples. A straightforward approach that directly utilizes LMNN will not be sufficient (see lines 159). It's also worth noticing that naively assigning all OOD samples to 1 class will not work (see our response to question 1 of reviewer 5hh7).
> > >   - Consequently, our method does not require retraining the feature extractor. Furthermore, with our novel modifications, our loss function is also inherently applicable in both inductive and transductive learning. To our best knowledge, POODLE is the first loss function that has this property.
> > >
> > > We will put this discussion in the revised paper.
> > >
> > > ### 3.  It would be nice if you can show that this approach is still useful, in the scaled up setting.
> > >
> > > Thanks for your recommendation, we have conducted experiments to integrate the pre-trained weights from other papers to our work below.
> > >
> > > TABLE 1: **Inductive** results of available pretrained of Resnet-12 on mini-Imagenet. Results are evaluated on 10,000 tasks.
> > >
> > > |                    | RFS [3] 1-shot | RFS [3] 5-shot | FEAT [4] 1-shot | FEAT [4] 5-shot |
> > > | ------------------ | -------------- | -------------- | --------------- | --------------- |
> > > | Pretrained         | 64.07±0.20     | 81.49±0.10     | 65.37±0.20      | 82.10±0.10      |
> > > | + with OOD samples | **64.91±0.20**     | 81.60±0.10     | **66.95±0.20**      | **82.36±0.10**     |
> > >
> > > TABLE 2: **Transductive** results of available pretrained of Resnet-12 on mini-Imagenet. Results are evaluated on 10,000 tasks.
> > >
> > > |                                | RFS [3] 1-shot | RFS [3] 5-shot | FEAT [4] 1-shot | FEAT [4] 5-shot |
> > > | ------------------------------ | -------------- | -------------- | --------------- | --------------- |
> > > | Pretrained (Inductive)         | 64.07±0.20     | 81.49±0.10     | 65.37±0.20      | 82.10±0.10      |
> > > | + with Pull Loss               | 73.42±0.24     | 83.95±0.13     | 72.99±0.24      | 84.12±0.13      |
> > > | + with Pull Loss + OOD samples | 75.96±0.24     | 84.30±0.13     | 75.30±0.24      | 84.51±0.13      |
> > >
> > > We can see that OOD samples could also enhance these pretrained models in both inductive & transductive settings.
> > >
> > > **References**
> > >
> > > [1]  Large Margin Few-Shot Learning
> > >
> > > [2] Distance Metric Learning for Large Margin Nearest Neighbor (JMLR 2009)
> > >
> > > [3] Rethinking Few-Shot Image Classification: a Good Embedding Is All You Need? (ECCV 2020)
> > >
> > > [4] Few-Shot Learning via Embedding Adaptation with Set-to-Set Functions (CVPR 2020)

---

> > > > ### Comment · Reviewer_C2g5 · 2021-09-01
> > > > **Thanks a lot for the additional clarifications.**
> > > >
> > > > Thanks a lot for the additional clarifications. I really appreciate it.
> > > >
> > > > I think incorporating all the discussions and new experiments that are provided during the rebuttal in the paper would make it much stronger and taking the rebuttal into account, I am increasing my score.
> > > >
> > > > P.S.: About point 3, how about running experiments with bigger models trained on larger datasets. E.g., would it be possible to apply your method on top of pretrained checkpoints like this one: https://huggingface.co/google/vit-large-patch32-384

---

### Official Review · Reviewer_5hh7 · 2021-07-17

**Rating:** 5
**Confidence:** 4

**Summary:**

This paper aims to address the problem of few-shot learning by exploiting unlabeled data from outside target novel classes. The main contribution is that it uses these out-of-distribution data as negative examples and learns a target classifier that simultaneously discriminates among the target classes and the negative class. To this end, the negative class is constructed by either sampling samples from base classes or from relevant classes of the target domain. A large-margin based prototype classifier is trained accordingly, which pulls together positive samples and pushes apart negative samples. A modified version of the classifier using stop-gradient operator is further introduced. The approach is tested in a variety of few-shot classification settings, including in-domain, cross-domain, and imbalanced distribution, and for both inductive and transductive scenarios, and compared with several baseline methods.

**Limitations And Societal Impact:**

It looks reasonable to me.

**Main Review:**

Originality:
1) The authors tackle an important and challenging problem of few-shot classification with out-of-distribution data. The proposed approach is simple and interesting.
2) The proposed approach can be viewed as learning a large-margin classifier on novel classes by using additional “negative” examples and with pre-trained feature extractor on base classes. There is a lack of discussion on relevant approaches that also focus on learning better classifiers on novel classes with pre-trained features. See the first point in the “quality” comments.

Quality:
1) While a series of experiments were conducted, the comparisons did not fully validate the proposed approach. In particular, the main comparisons focused on baselines that train the feature extractors in different ways (e.g., with standard CE loss, with additional rotation self-supervision and with knowledge distillation). While these comparisons did show that the proposed approach works with different types of pre-trained features, to me these are just one aspect of the comparison. What were supposed to be the main comparisons would be comparing against approaches that also focus on learning better classifiers on novel classes with pre-trained features. There are different strategies: for example, (a) learning cosine classifiers as in [3]; (b) learning large-margin classifiers without out additional negative examples; (c) refining few-shot classifiers as in [Learning to learn: Model regression networks for easy small sample learning, ECCV, 2016]; (d) performing data augmentation at test time as in [Free lunch for few-shot learning: Distribution calibration, ICLR, 2021]; etc. Are the proposed approach better than these existing works?
2) The proposed approach uses large-margin classifiers to pull positive samples and push negative samples. An alternative is contrastive learning, which is widely used recently. How is the connection, difference, and comparison between these two?
3) How does the size of negative examples K affect the performance? In Table 10 of the supplementary material, it shows that K is helpful when K=500. Does the performance further improve, saturate, or drop with larger K?
4) Table 3 and Table 4 miss state-of-the-art approaches. Also, from https://paperswithcode.com/dataset/miniimagenet-1 etc., the proposed approach is worse than state of the art.
5) The authors mentioned that the proposed approach leads to a little overhead at inference. It would be useful to provide more quantitative analysis, like comparing the classifier training time.

Clarity:
1) The paper generally reads well.
2) Some important technical details are not clear. For example, the stop gradient operator is critical in this paper (Line 162) but without enough details; what is the difference between POODLE-B and POODLE -R (Table 1 caption). While some of the explanations are provided in the supplementary material, I feel they should be discussed in the main paper.

Significance:
1) Experimental evaluations demonstrate the effect by introducing the out-of-distribution data as negative examples.
2) The proposed approach assumes available out-of-distribution examples. This is sometime unrealistic in practice for cross-domain settings, as mentioned by the authors as well (Line 175-181). The authors considered disjoint and noisy negatives for cross-domain settings. A more realistic scenario is in fact to directly use examples from base classes as negative. Does this work?
3) If we consider the performance in joint base and novel label space which is important in some real-world applications, is the proposed approach still effective?
4) Following the standard practice in few-shot evaluation, it would be helpful to include the 95% confidence intervals in the tables.

My initial rating is based on the comments above. I think the paper can be substantially improved by a addressing these comments.

Post Rebuttal:

I do appreciate the efforts and additional experiments that the authors made in the rebuttal. While this paper proposed an interesting setting/approach to few-shot classification, the technical novelty is somehow limited regarding the use of large margin classifiers for few-shot classification, as mentioned by other reviewers as well. From the experimental comparisons in the original submission and in the rebuttal, the proposed approach is comparable to and does not show significant improvements over the state-of-the-art methods, such as the distribution calibration paper. But this approach requires additional examples from related domains, while the distribution calibration paper does not. Also, such reliance on additional examples from related domains might make the approach inapplicable in practice, where this kind of data is difficult to acquire. In addition, some important details are vague and unclear, which is not fully clarified in the rebuttal. Some of these concerns are addressed in the rebuttal, but not fully clarified. This makes the paper not ready for this NeurIPS. The presentation and the writing flow can be improved as well. I encourage the authors to continue this line of work for future submission.

**Time Spent Reviewing:**

3

---

> ### Author Response · Authors · 2021-08-10
> **Thanks for your insightful review.**
>
> ### 1. Comparisons against approaches for learning better classifiers on novel classes with pre-trained features.
>
> We thank the reviewer for recommending more comparisons in this aspect. As suggested, we added the following methods for comparison: 1) A baseline with cosine distance loss; 2) The same baseline 1 with learning cosine classifier approach; 3) Baseline 1 with naive OOD where we train the linear classifier with 1 class for OOD samples and normal CE loss; 4) Baseline 1 with a large-margin classifier w/o negative samples; 5) The Free-lunch method [2].
>
> For the Free-lunch method, we report the number from the original paper. In this experiment, we use the *rot + WRN-28-10 pretrained* setting and evaluate with 10,000 tasks. Note that our baseline is comparable to Free-lunch, which also uses WRN-28-10 + Rotation loss.  The result is shown in the inline table.
>
>
> | Method                            | 1-shot          | 5-shot          |
> | --------------------------------- | --------------- | --------------- |
> | 1) Baseline                          | 64.77±0.20     | 83.66±0.13     |
> | 2) w/ learning cosine classifier        | 65.46±0.20     | 83.66±0.13     |
> | 3) w/ naive OOD                         | 65.36±0.20   | 83.08±0.13     |
> | 4) w/ large-margin w/o negative samples | 64.77±0.20     | 83.66±0.13     |
> | 5) Free-lunch [2]                        | **68.57±0.55** | 82.88±0.42     |
> | 6) POODLE-B                          | **68.40±0.20** | **84.49±0.13** |
>
> Our observation is as follows.
>
> 2\) Learning cosine classifier with only CE loss usually does not bring meaningful improvement. In our experience, the enhancement of baseline++ over baseline in [3] is *mainly because of the l2-normalization* not the fine-tuning procedure itself. In our experiment, removing the l2-norm deteriorates the performance of the network by approximately 5-10% in 1 shot protocol depending on the pretrained models. This result is also consistent with the finding in [1].
>
> 3\) Naive OOD performs worse than POODLE because the OOD samples are drawn from several classes of the training set, which are well-clustered and separated, we cannot find a "prototype" with a linear classifier to match all of those negative samples.
>
> 4\) Large-margin w/o negative samples (inductive) is not better than cosine-distance because the initialized prototype (mean of all samples from specific class) is already well-clustered and optimal i.e., close to the ground-truth class prototype and far away from others **for the observed samples**.
>
> 5\) Free-lunch: We note that in their paper, the authors claim to improve baseline around 10% in the 1-shot protocol, however, it can be misleading since they do not use l2-normalization in their baseline, thus, dropping the baseline's performance significantly. We reran **their pretrained WRN + rotation loss** with l2-normalization + cosine classifier (without any other tweaks) on mini-Imagenet and achieved **65.36±0.20/83.11±0.13** for 1-shot/5-shot protocol compared to 56.37±0.68/79.03±0.51 of reported unnormalized baseline. Hence, *the performance gain is roughly 3%* compared to baseline (also found in issue#3 of official code [2]), which is approximately equal to POODLE. Furthermore, *their approach does not bring improvement in the 5-shot protocol as ours* (also noted by the author in issue#3). Without l2-normalization, POODLE also "boost" the un-normalized baseline by 5-10%.
>
> ### 2. The connection between the large-margin classifier approach and contrastive learning.
>
> In contrastive learning, a negative pair is formed by two original samples in the training data, and a positive pair is formed by an original sample and its augmentation. Intuitively, it is not guaranteed that semantically similar samples stay close to each other in the embedding due to the repelling effect on negative pairs. It is possible to apply the large-margin principle to regularize the inter-class and intra-class distances in contrastive learning, as shown by Chen et al., Large-Margin Contrastive Learning with Distance Polarization Regularizer, ICML 2021. Extending our work with the contrastive loss would be interesting future work.
>
>
> ### 3. How does the size of negative examples K affect the performance? In Table 10 of the supplementary material, it shows that K is helpful when K=500. Does the performance further improve, saturate, or drop with larger K?
>
> We have conducted more experiments and reported the result with different K below.
> The result demonstrates that performance gain saturates at about 400-500 samples.
>
> | Number of negative samples | 1-shot     | 5-shot     |
> | -------------------------- | -----------| -----------|
> | 0                          | 66.32±0.20 | 82.99±0.13 |
> | 10                         | 67.05±0.20 | 83.20±0.13 |
> | 50                         | 67.61±0.20 | 83.56±0.13 |
> | 100                        | 67.73±0.20 | 83.63±0.13 |
> | 200                        | 67.78±0.20 | 83.68±0.13 |
> | 400                        | 67.84±0.20 | 83.72±0.13 |
> | 500                        | 67.83±0.20 | 83.73±0.13 |
> | 750                        | 67.84±0.20 | 83.74±0.13 |
> | 1000                       | 67.84±0.20 | 83.73±0.13 |
>
>
>
> ### 4. Table 3 and Table 4 miss state-of-the-art approaches. Also, from https://paperswithcode.com/dataset/miniimagenet-1, etc., the proposed approach is worse than state-of-the-art.
>
> Please see our response to common issues. Leaderboard results reported by paperswithcode should be taken with a grain of salt because their experiment setting could be unfair for direct comparisons with results reported in the papers. Details of the algorithm are often lacking or omitted in the leaderboard. For example, it is well-known that large architectures such as DenseNet, WRN, etc. usually lead to significantly higher results, while we use ResNet-12 in our experiments. The experimental results of our approach on different backbones can be found in the supplementary document. For few-shot learning, the leaderboard does not clearly indicate whether the transductive or inductive setting is used. The transductive setting can often have remarkable improvement compared to the inductive one. Some methods even used semi-supervised learning or pretraining on large datasets such as ImageNet-1000. Last but not least, some entries on paperswithcode haven't gone through peer-reviewed.
>
> However, we will revise Tables 3 and 4 with the updated comparisons as done in the tables in the common issues response.
>
> ### 5. The authors mentioned that the proposed approach leads to a little overhead at inference. It would be useful to provide more quantitative analysis, like comparing the classifier training time.
>
> We have included the inference time of Resnet-12 with 1-shot inductive protocol on mini-Imagenet here. All approaches are evaluated on Nvidia DGX-A100 system and used 1 GPU. Please see Section 5.1 for hyper-parameters setup. The results are shown in the inline table. As can be seen, our method takes about 15 seconds to perform inference on 10,000 episodes.
>
> | Method          | Time for inference 10,000 episodes (sec) |
> | --------------- | ------------------------------------- |
> | 1) Baseline | 0.5694                                |
> | 2) w/ learning cosine classifier | 15.0501                               |
> | 6) POODLE     | 15.1073                               |
>
>
> ### 6. The authors considered disjoint and noisy negatives for cross-domain settings. A more realistic scenario is in fact to directly use examples from base classes as negative. Does this work?
>
> The reviewer's question is that why we did not use samples from base training data (i.e., the training set of mini-Imagenet in this case) when testing on the cross-domain dataset (e.g., CUB, EuroSAT). Please clarify if it is wrong.
>
> It is expected that the negative samples should have meaningful cues to effectively refine the classifier. In cross-domain FSL,  using examples from base classes as negative examples will result in large discrepancies between positive and negative samples, therefore, does not benefit the classifier. Consequently, we need to use negative samples from the same domain with test data e.g., using EuroSAT OOD samples when testing on EuroSAT. Please see our answer to Question 2 of Reviewer C2g5 for more details.
>
> It is worth noticing that other methods in Table 4 also used 20% of unlabeled data from the same domain of test data, hence, the comparison is fair.
>
>
>
> ### 7. Following the standard practice in few-shot evaluation, it would be helpful to include the 95% confidence intervals in the tables.
>
> We have reported the results with 95% confidence intervals of Resnet-12 on mini-Imagenet in common response. The half-width of other tables should be roughly 0.4/0.2 for 1-shot/5-shots when evaluating 2,000 episodes, indicating the improvement from POODLE is statistically significant. We will amend this in the revised paper.
>
> **References**
>
>    [1] SimpleShot: Revisiting Nearest-Neighbor Classification for Few-Shot Learning
>
>    [2] Free Lunch for Few-Shot Learning: Distribution Calibration (ICLR 2021) - official code: https://github.com/ShuoYang-1998/Few_Shot_Distribution_Calibration
>
>    [3] A Closer Look at Few-shot Classification (ICLR 2019)

---

> ### Author Response · Authors · 2021-09-02
> **Clarify for concerns after rebuttal**
>
> Thanks for your response. Though the reviewer has made the decision, we find it would be better to clarify some of your concerns below.
>
> ### 1. The technical novelty is somehow limited regarding the use of large margin classifiers for few-shot classification, as mentioned by other reviewers as well.
>
> We would like to clarify that studying the large margin principle is **not** the target of this paper. Specifically, the key contribution of our paper is the idea of capitalizing OOD samples to tackle the ambiguity of FSL (also see the first answer of response to common issues). We believe this concept is novel and the novelty of this work can be considered as to how we assembled the well-known techniques i.e., margin loss to tackle this problem. Also, the margin loss is not straightforwardly applicable but requires several (novel) modifications (see lines 159). We also have a different perspective for utilizing LMNN than prior work, which we articulated in [response to question 2 of reviewer C2g5](https://openreview.net/forum?id=sfzseGUqFrd&noteId=IpMWfP05PTG).
>
> ### 2. The proposed approach is comparable to and does not show significant improvements over the state-of-the-art methods, such as the distribution calibration paper. But this approach requires additional examples from related domains, while the distribution calibration paper does not.
>
> We would like to point out that:
>
> - For in-domain FSL (since [1] only report results in this setup)
>
>   - On miniImagenet, our approach does better than WRN in 5-shot inductive learning. Our results on Resnet-12 also outperforms Resnet-18 in [1] (which has comparable architecture and number of parameters). Also, we believe that aforementioned papers have their own merits besides the number on the leaderboard.
>   - For in-domain, the OOD samples are **readily available** from base data (e.g., sampled from 38400 images in case of mini-Imagenet) for training the feature extractor - which to our point of view is reasonable.
>   - Furthermore, if we don't have access to the base training data, for in-domain, we can use the random features to approximate the trained data i.e., POODLE-R (see our [response to questions 1 & 2 of reviewer C2g5](https://openreview.net/forum?id=sfzseGUqFrd&noteId=PXWvcLF4_aa) for the elaboration of why it works).
>
>   Thus, **we do not need any additional data** for the reported setup of [1].
>
> - For cross-domain FSL:
>
>   - We demonstrate the effectiveness of our method in extreme cross-domain FSL (and transductive learning), which is not available in [1].
>   - Compared to other work that also tackle (extreme) cross-domain FSL (such as [2]), our approach is **more efficient** because we can reuse 400  *unlabeled* samples for all test episodes. In contrast, [2] requires 20% unlabeled data from dataset (~ 48600 images for iNat, 5400 images for EuroSAT, 2000 images for ISIC, 1200 images for CUB).
>
> ### 3. In addition, some important details are vague and unclear, which is not fully clarified in the rebuttal.
>
> Regarding your concern for clarity, we have presented the analysis for the stop-gradient in Section B of supplementary. The requested half-width of 95CI is presented in our response to question 7 of the reviewer. We clarify the difference between POODLE-R and POODLE-B in response to reviewer C2g5 and will revise the paper accordingly.
>
> The clarity issues raised by reviewer C2g5 (why random features works) and FGzB (how to tune hyperparameters) are addressed in our response to their review, respectively.
>
> Lastly, all the hyper-parameters are reported in Section 5.1.
>
> **References**:
>
> [1] Free Lunch for Few-shot Learning: Distribution Calibration (ICLR 2021)
>
> [2] Self-training For Few-shot Transfer Across Extreme Task Differences (ICLR 2021)

---

### Author Response · Authors · 2021-08-10
**Response to common issues**

We are grateful to the reviewers for providing constructive feedback. The reviewers appreciated our work on an important and challenging problem of few-shot classification and that the proposed approach is simple and interesting (Reviewer 5hh7). Our method is applicable on top of other techniques to improve FSL performance, and the results show significant improvements over baselines based on the numbers reported in the paper (Reviewer C2g5). The proposed model is simple but effective, and negative samples are easy to obtain (Reviewer FMo7). The paper is well-written and presents an interesting idea and direction (Reviewer FGzB). There are some common concerns raised by the reviewers, which we addressed here before responding to individual comments.

### 1. Contribution and Novelty

Our paper presents a method for leveraging out-of-distribution (OOD) samples to reduce ambiguity in few-shot learning. We believe that this concept is novel in the context of few-shot learning (as also acknowledged by Reviewer FMo7), which has not been explored before.
There are many possibilities to implement this concept, wherein this paper we approach by using the large-margin principle. However, marrying large-margin nearest neighbors and OOD is a non-trivial task; it requires some crucial modifications in making the implementation to achieve state-of-the-art performance e.g., soft-weighted class assignment and stop-gradient operator. Please see more details in Table 9.

We believe that such technical contributions and our SOTA results would benefit the community and advance the subject of few-shot learning. We will revise the text to emphasize these contributions.

### 2. Comparison with SOTA approaches

As suggested by the reviewers, we include comparisons of our results with more recent works in Table 1 and 2, which is for Resnet-12 and WRN-28-10 backbone, respectively. The results of other methods are taken from their original papers. **As can be seen, our method outperforms the SOTA methods**.

In both tables, we also provide additional comments on the implementation details used for generating the results. In the tables, SSL and KD denote self-supervised learning and knowledge distillation. *It's worth noticing that, using episodic training, SSL, and KD should be equivalent in terms of baseline enhancement (roughly 2-3% on average for 1-shot protocol)*.


TABLE 1: **Inductive** mini-Imagenet with **Resnet-12** backbone. Our results are evaluated with **10,000** episodes with the same configuration as specified in the paper (see section 5.1).


| Method                      | 1-shot          | 5-shot          | use SSL | use episodic training | Additional comments          |
| --------------------------- | --------------- | --------------- | :-------: | :----------: | ---------------------------- |
| DSN [1]                     | 62.64±0.66     | 78.83±0.45     |         |                       |                              |
| MetaOptNet [2]              | 62.64±0.61     | 78.63±0.46     |         |                       |                              |
| LR+DC [3]                   | 61.50±0.47     | N/A             |         |                       | resnet-18                    |
| **POODLE-B + simple** | **64.08±0.20** | **81.41±0.13** |         |                       |                              |
| FEAT [4]                    | 66.78±0.20     | 82.05±0.14     |         | ✓                     | additional params            |
| ArL  [5]       | 65.21±0.58     | 80.41±0.49     |         |                       | use semi-supervised learning |
| FRN [6]                     | 66.45±0.19     | 82.83±0.13     |         | ✓                     |                              |
| DeepEMD [7]                  | 65.91±0.82     | 82.41±0.56     |         | ✓                     |                              |
| PSST [8] | 64.05±0.49     | 80.24±0.45     | ✓       |                       |                              |
| **POODLE-B + rot**          | **67.20±0.20** | **83.74±0.13** | ✓       |                       |                              |
| IEPT [9] | 67.05±0.44 |  82.90±0.30 | ✓       |        ✓          |
| **POODLE-B + rot + kd**     | **67.84±0.20** | **83.72±0.13** | ✓       |                       | use KD                       |

TABLE 2: **Inductive** mini-Imagenet with **WRN-28-10** backbone. Our results are evaluated with **10,000** episodes with the same configuration as specified in the paper (see section 5.1).

| Method                  | 1-shot          | 5-shot          | use SSL | use episodic training | Additional comments             |
| ----------------------- | --------------- | --------------- | :-------: | :---------: | ------------------------------- |
| LEO [10]                 | 61.76±0.08     | 77.59±0.12     |         |                       |                                 |
| AWGIM [11]               | 63.12±0.08     | 78.40±0.11     |         |                       |  |
| LR+DC [3]               | **64.38±0.63** | N/A             |         |                       |                                 |
| **POODLE-B + simple**   | **64.97±0.20** | **81.87±0.13** |         |                       |                                 |
| FEAT [4]                | 65.10±0.20     | 81.11±0.14     |         | ✓                     | auxiliary param                 |
| PSST [8]                | 64.16±0.44     | 80.64±0.32     | ✓       |                       |                                 |
| LR+DC [3]               | **68.57±0.55** | 82.88±0.42     | ✓       |                       |                                 |
| **POODLE-B + rot**      | **68.40±0.20** | **84.49±0.13** | ✓       |                       |                                 |
| POODLE-B + rot + kd | 69.68±0.20 | 84.82±0.13 | ✓       |                       | use KD                          |

**References**

[1] Adaptive Subspaces for Few-Shot Learning (CVPR 2020)

[2] Meta-Learning with Differentiable Convex Optimization (CVPR 2019)

[3] Free Lunch for Few-shot Learning: Distribution Calibration (ICLR 2021)

[4] Few-Shot Learning via Embedding Adaptation with Set-to-Set Functions (CVPR 2020)

[5] Rethinking Class Relations: Absolute-relative Supervised and Unsupervised Few-shot Learning (CVPR 2021)

[6] Few-Shot Classification with Feature Map Reconstruction Networks (CVPR 2021)

[7] DeepEMD: Differentiable Earth Mover's Distance for Few-Shot Learning (CVPR 2020)

[8] Pareto Self-Supervised Training for Few-Shot Learning (CVPR 2021)

[9] IEPT: Instance-Level and Episode-Level Pretext Tasks for Few-Shot Learning   (ICLR 2021)

[10] Meta-Learning with Latent Embedding Optimization (ICLR 2018)

[11] Attentive Weights Generation for Few Shot Learning via Information Maximization (CVPR 2020)

---

### Decision · Program_Chairs · 2021-09-28

**Decision:**

Accept (Poster)

**Comment:**

This paper proposes using unlabeled samples outside of target classes to improve few-shot learning. The main idea is that discriminating these samples from in-distribution samples would allow the model to improve feature learning.

Reviewers recognize that the proposed method improves over SOTA but they point to limited technical novelty regarding the use of large margin, marginal improvements over SOTA methods, the reliance of methods on extra data as the shortcomings of this work. However, all reviewers believe that after improving the presentation of the paper and the empirical results, this paper would make a great publication.

Given the above concerns, I recommend rejecting the paper and resubmitting it after taking reviewers' suggestions into account.

**Consistency Experiment:**

NeurIPS has a long history of experimentation. In 2014, NeurIPS ran an experiment in which 10% of submissions were reviewed by two independent committees to quantify the randomness in the review process. This year, we repeated a variant of this experiment to see how the quality of the review process has changed over time.  This paper was part of the experiment and was therefore assigned to two committees (consisting of reviewers, an Area Chair, and a Senior Area Chair) that reached independent decisions.  If both committees made the same recommendation, this recommendation was followed. If a single committee recommended acceptance, the paper was accepted (with the exception of a few cases in which the other committee identified what we considered a fatal flaw, e.g., an error in a key result).

This copy’s committee reached the following decision: **Reject**

The other committee assigned to the paper recommended **Accept (Poster)**.  You can find the other set of reviews, along with any follow up discussion with the authors here:
https://openreview.net/forum?id=wEvO8BCqZcm